# Cerebellar Purkinje cells combine sensory and motor information to predict the sensory consequences of active self-motion in macaques

**Omid A. Zobeiri[1] & Kathleen E. Cullen [2,3,4,5]** ✉

Accurate perception and behavior rely on distinguishing sensory signals arising from unexpected events from those originating from our own voluntary actions. In the vestibular system, sensory input that is the consequence of active self-motion is canceled early at the first central stage of processing to ensure postural and perceptual stability. However, the source of the required cancellation signal was unknown. Here, we show that the cerebellum combines sensory and motor-related information to predict the sensory consequences of active self-motion. Recordings during attempted but unrealized head movements in two male rhesus monkeys, revealed that the motor-related signals encoded by anterior vermis Purkinje cells explain their altered sensitivity to active versus passive self-motion. Further, a model combining responses from ~40 Purkinje cells accounted for the cancellation observed in early vestibular pathways. These findings establish how cerebellar Purkinje cells predict sensory outcomes of self-movements, resolving a long-standing issue of sensory signal suppression during self-motion.

As we interact with the world, our sensory systems encounter stimuli that are the consequences of our own actions. The brain's ability to dynamically predict the sensory consequences of our actions is essential for both accurate motor control and perceptual stability. For example, we can readily distinguish whether our self-motion is intended or the result of an unexpected event such as slipping on ice, and in turn naturally make appropriate motor responses. The prevailing view is that the brain makes this vital distinction by computing a prediction of the sensory consequences of action based on internally generated motor signals using learned internal models. The resulting prediction is then compared to the incoming sensory signal to attenuate neural responses to expected stimuli such that neural responses preferentially encode novel or unexpected stimuli.

During active behaviors, self-generated vestibular signals are encoded in a context-independent manner by the vestibular afferents of the VIII nerve, such that their responses are comparable to those evoked by unexpected vestibular stimuli[1–4]. In contrast, vestibulospinal pathway neurons in the vestibular nuclei that are directly targeted by these afferents as well as neurons in the deep cerebellar nuclei (i.e., rostral fastigial nucleus which is reciprocally connected to vestibular nuclei[5]), demonstrate comparable cancellation for reafferent vestibular inputs during active versus passive head movements (~70%[6–11]), as do neurons in the ascending posterior thalamocortical vestibular pathway[12]. In turn, this selective encoding of unexpected motion provides a neural mechanism for ensuring postural and perceptual stability during self-motion. However, the source of the suppression signal that cancels peripheral vestibular input to central

[1]Department of Biomedical Engineering, McGill University, Montréal, QC, Canada. [2]Department of Biomedical Engineering, Johns Hopkins University, Baltimore, MD, USA. [3]Department of Otolaryngology-Head and Neck Surgery, Johns Hopkins University School of Medicine, Baltimore, MD, USA. [4]Department of Neuroscience, Johns Hopkins University School of Medicine, Baltimore, MD, USA. [5]Kavli Neuroscience Discovery Institute, Johns Hopkins University, Baltimore, MD, USA. ✉e-mail: Kathleen.Cullen@jhu.edu

pathways during expected self-motion remains unknown. Notably, given that vestibular nuclei neurons can explicitly distinguish expected from unexpected vestibular stimuli even when experienced simultaneously[11–14], there is no evidence for presynaptic inhibition of the afferent central neuron synapse (reviewed by ref. 15). Taken together, then, these findings raise the question: what is the central source of the sensory suppression signal required to cancel peripheral vestibular input and how is it computed?

There are many reasons to believe that the cerebellum plays an essential role in generating the required suppression signal. First, patients with cerebellar lesions demonstrate impairments in the ability to accurately estimate the sensory consequences of motor commands[16–18]. Second, experiments disrupting cerebellar activity in healthy subjects via transcranial magnetic and direct current stimulation[19–21] have provided causal evidence for the cerebellum's role in predicting the sensory consequences of action. And third, this view is consistent with results of functional imaging studies that have provided evidence for the coding of unexpected sensory signals in the cerebellar BOLD response[22–24]. More specifically, in the context of self-motion processing, the anterior vermis of the cerebellar cortex performs a fundamental role in facilitating the coordination of posture and active movements. Anterior vermis Purkinje cells send strong direct inhibitory inputs to both the vestibular and deep cerebellar nuclei[25,26]. Furthermore, in response to unexpected self-motion, these Purkinje cells integrate vestibular and proprioceptive stimuli[27,28] to mediate the transformation of vestibular information into the body-centric reference frame required for accurate postural control[28]. To date, however, it is currently unknown how these neurons respond when self-motion is actively generated and thus expected.

Accordingly, here we recorded from individual Purkinje cells in the anterior vermis of the cerebellar cortex, while monkeys made active head movements. Specifically, the goal of our present study was to test the hypothesis that Purkinje cells are the source of the suppression signal to the vestibular and deep cerebellar nuclei during expected self-motion. First, our results reveal that anterior vermis Purkinje cells differentially encode active versus passive movements in a manner consistent with their motor-related inputs. This contrasts with rFN and VN neurons, which show no modulation to the generation of head motor commands. Second, using a simple population-based model, we then demonstrate that the convergence of the signal transmitted by ~40 Purkinje cells accounts for the suppression of self-generated vestibular signals in their target neurons in the vestibular and deep cerebellar nuclei. Taken together our results constitute the direct demonstration of the widely held view that the cerebellum integrates movement-based predictions with actual sensory feedback to cancel the sensory consequences of active self-motion.

## Results

All Purkinje cells included in this study ($n = 63$) were sensitive to passive vestibular stimulation and were insensitive to eye movements. To assess each neuron's vestibular sensitivity, we applied ipsilaterally and contralaterally directed whole-body rotations in the dark (i.e., whole-body rotations; see "Methods"). Neurons were then further characterized as either unimodal or bimodal based on their simple spike responses to passively applied stimulation of neck proprioceptors. Correspondingly, to assess each neuron's proprioceptive sensitivity, we applied ipsilaterally and contralaterally directed rotations to the monkey's body while its head was held stationary relative to space (i.e., body-under-head rotations; Supplementary Fig. 1, see Methods) using the same motion profiles as those used for the assessment of vestibular sensitivities. The majority of Purkinje cells in our population (70%) were responsive both to passive proprioceptive and vestibular stimulation. These neurons were therefore classified as bimodal Purkinje cells. The remaining 30% of cells in our population were responsive to vestibular but insensitive to proprioceptive stimulation

and were therefore classified as unimodal Purkinje cells. Overall, our population of anterior vermis Purkinje cells responded to passive vestibular and proprioceptive stimulation in a manner consistent with previous characterizations of these neurons in the rhesus monkey[28].

## Purkinje cell simple spike responses are suppressed for active relative to passive vestibular stimulation

In everyday life, vestibular and proprioceptive inputs are generally the result of our own actions. To date, however, prior investigations have exclusively focused on understanding how anterior vermis Purkinje cells integrate vestibular and proprioceptive sensory information resulting from unexpected self-motion such as the passive stimulation conditions described above[27,28]. This then raises the question: how these same Purkinje cells respond to self-motion when it is instead actively generated and thus expected.

To directly address this question, we first compared Purkinje cell simple spike responses to passively applied and self-generated head motion stimuli with comparable profiles. Neuronal responses were initially recorded during passively applied vestibular stimulation characterized by a velocity profile designed to mimic the profile produced during active movements (i.e., "active-like" motion profiles; see Experimental Procedures). The same passive rotations were then applied with the head held stationary relative to space to assess responses to comparable proprioceptive stimulation. Quantification of these vestibular and proprioceptive sensory responses (Supplementary Fig. 2) demonstrated sensitivities that were consistent with prior characterizations of vestibular-sensitive anterior vermis Purkinje cells[28]. Neuronal responses were then recorded during a third passive "active-like" motion protocol in which the monkey's head was rotated on its stationary body to produce simultaneous vestibular and proprioceptive stimulation.

We then released the monkey's head to allow the generation of active head-on-body movements. Figure 1 illustrates the marked difference in the responses of two representative Purkinje cells during comparable passive (Fig. 1a) versus active (Fig. 1b) head-on-body movements. The response of each Purkinje cell in our population was quantified using a least-squares dynamic regression model with three kinematic terms (i.e., head-in-space position, velocity, and acceleration) (Supplementary Fig. 3, see Materials and methods). The example unimodal and bimodal Purkinje cells both displayed strong modulation in the passive head motion condition. In contrast, both example neurons were far less responsive to the same motion when it was actively generated (60% and 75% reduction in response, respectively). Furthermore, the example bimodal Purkinje cell's response was not only suppressed in the active condition, but its modulation was in the opposite direction to that observed during comparable passive head motion (i.e., reduced rather increased modulation; Fig. 1b, red arrow).

The observations shown in Fig. 1 are summarized for our population of Purkinje cells in Fig. 2. Overall, both bimodal and unimodal neurons (Fig. 2, filled and open bars, respectively) showed a marked reduction in their modulation for active head movements in the preferred direction with respect to their responses in the passive condition (~92% $p < 0.001$). Comparable results were also found for active head movements in each neuron's nonpreferred direction (Supplementary Fig. 4, 90%, $p < 0.001$, and 86%, $p = 0.12$ respectively). Strikingly, we further found that the response direction of a significant population of Purkinje cells (~40%) actually reversed in the active relative to the passive condition. This result is illustrated by the distribution illustrated in Fig. 2b, where the sensitivities of individual neuronal in the active condition were frequently negative, indicating a reduction in neural modulation rather than the increase observed in the passive condition (Fig. 2a). Comparable results were also found in the nonpreferred direction, with 35% of neurons showing oppositely directed modulation in the active versus passive condition (Supplementary Fig. 4A versus B). Figure 2c plots the active versus passive

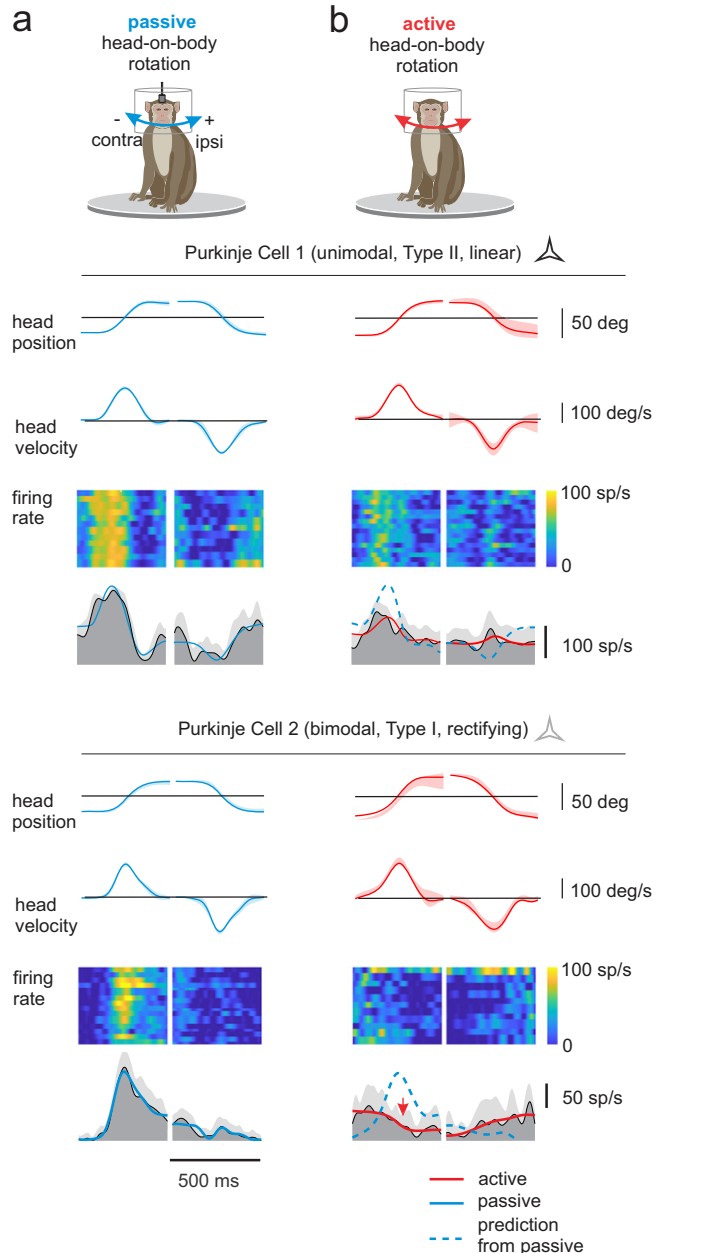

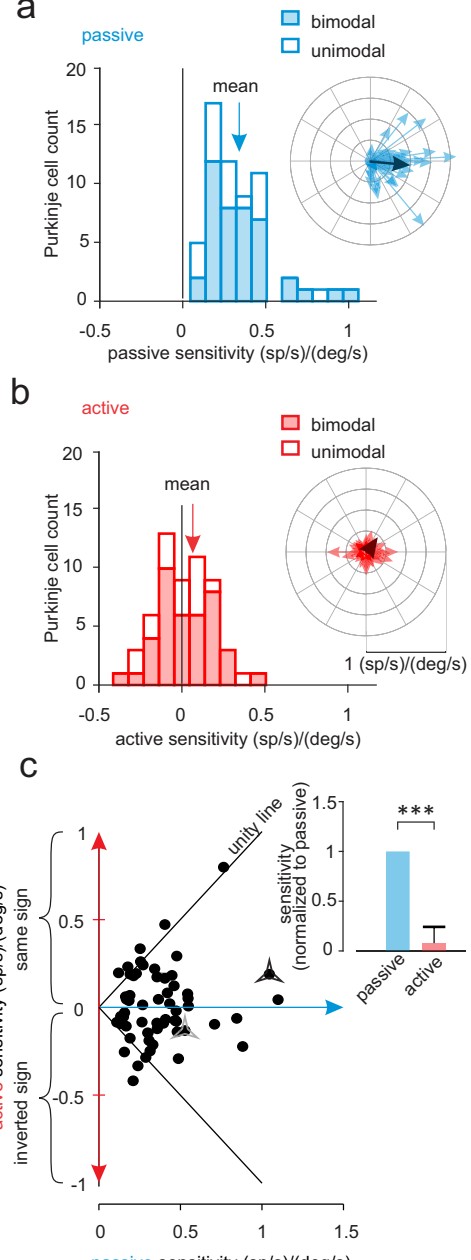

**Fig. 1 | Responses of example unimodal and bimodal (black and gray stars) Purkinje cells during passive.** (**a** blue traces) and active (**b** red traces) motion paradigms. Top two rows: The rotational head position and velocity for 15 movements presented as mean values ± SD. Heat map plots illustrating the example neurons' responses for each trial are shown below. Bottom row: Dark and lighter gray shading correspond to the average firing rates and standard deviations for the same 15 movements. Overlaying blue (**a**) and red (**b**) lines show the estimated best fit to the firing rate based on a bias term and sensitivity to passive and active head motion, respectively. Superimposed dashed blue lines in the active condition (**b**) show predicted responses based on each neuron's sensitivity to passive motion. Note, both neurons showed robust modulation for passive motion, but (i) responses were minimal when the same motion was self-produced and (ii) the response of the example bimodal Purkinje cell reversed direction in the active compared to passive condition (red arrow).

**Fig. 2 | Comparison of Purkinje cell responses to active versus passive head motion.** **a** Distribution of the neural sensitivities for our population of bimodal (filled bars) and unimodal (open bars) Purkinje cells during passive self-motion in the preferred direction (i.e., the direction resulting in the larger increase in simple spike firing rate). **b** Same as in (**a**) for active self-motion. *Insets:* polar plots where the vector length and angle represent each neuron's vestibular response sensitivity and phase, respectively. Dark arrows represent the population's average response. **c** Scatter plot showing a cell-by-cell comparison of response sensitivities to active and passive motion. The black solid lines represent unity and stars represent example Purkinje cells in Fig. 1. Insets: Bar plots comparing the normalized sensitivities of bimodal and unimodal Purkinje cells to passive vs. active head motion ($n = 63$, ***$p < 0.001$).

sensitivities for our population of Purkinje cells on a cell-by-cell basis. Notably, neuronal sensitives were generally less in the active than passive condition, even for those neurons that reversed direction (i.e., most points fall between the two unity lines). Furthermore, the dynamics of responses varied considerably across Purkinje cells in

each of two conditions (Supplementary Fig. 5). Overall, when considered as a population, we found that Purkinje cell response sensitivities were reduced by >80% when head motion was self-produced rather than externally applied (preferred direction, Fig. 2c inset, $p < 0.001$; nonpreferred direction, Supplementary Fig. 4C inset, $p < 0.001$).

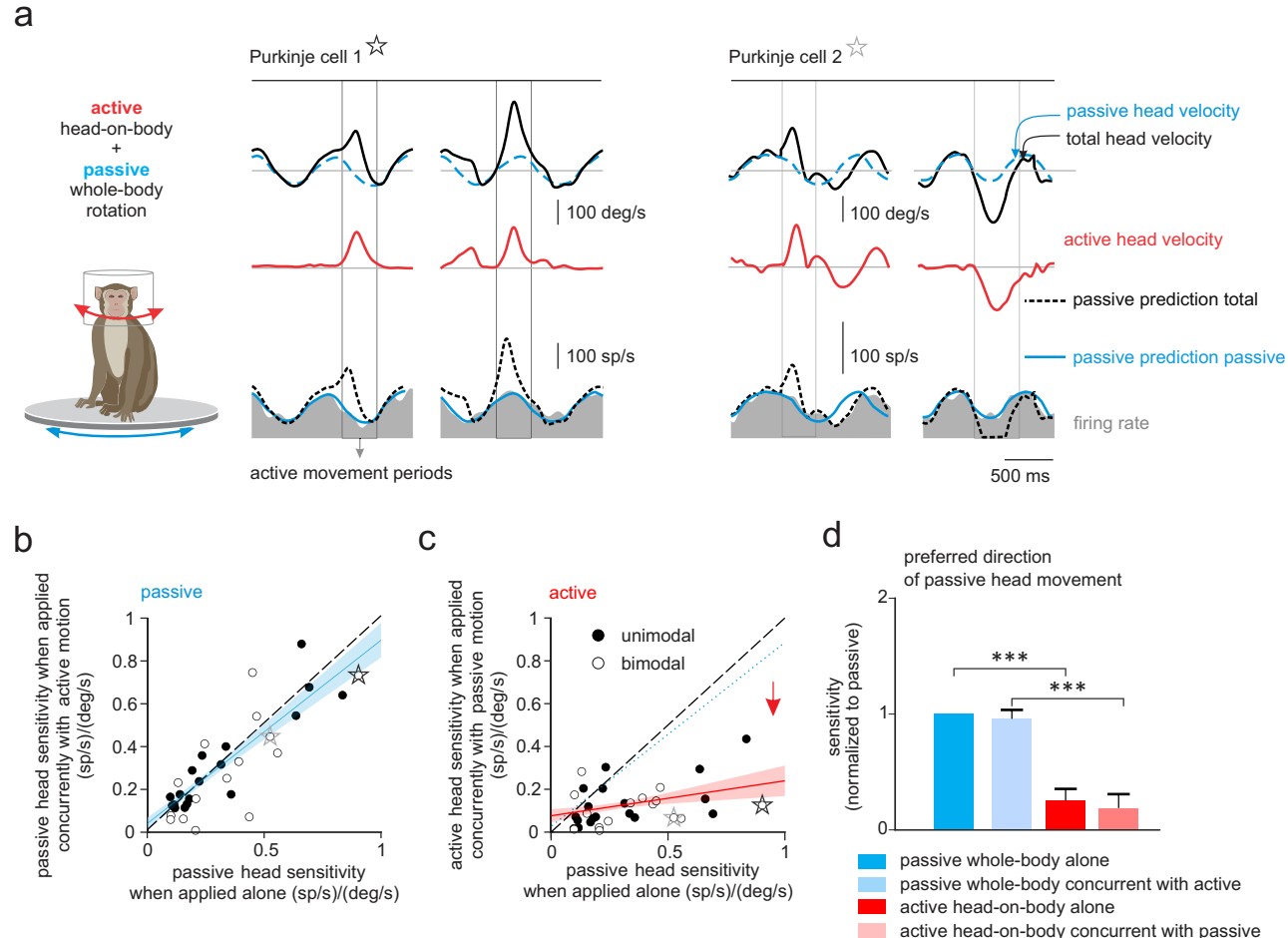

**Fig. 3 | Purkinje cells selectively encode passive head motion during simultaneously occurring active and passive rotation. a** Activity of the example unimodal and bimodal (black and gray stars, respectively). Purkinje cells during a paradigm where the monkey generated voluntary head movements while being simultaneously passively rotated. *Top and middle rows:* Total, passive, and active head velocities are shown (black, dashed blue, and red traces, respectively). *Bottom row*: Firing rates are shown in gray with superimposed traces illustrating firing rate predictions based on each neuron's sensitivity to passive head motion only (blue traces) versus total head motion (dashed black traces). The velocity traces for Purkinje cell 1 is illustrated with the opposite convention for the visual purpose. **b** Scatter plot of unimodal (black circles) and bimodal (open circles) Purkinje cell sensitivities to the passive component of simultaneously occurring active and passive rotation, versus their sensitivities to passive rotations occurring alone. The sensitivity of Purkinje neurons to passive motion was the same regardless of whether it occurred in isolation or in combination with active movements. **c** Scatter plot of unimodal and bimodal Purkinje cell sensitivities to the active component of simultaneously active and passive rotation versus sensitivities to passive rotations occurring alone. Blue (**b**) and red (**c**) lines and shading denote the mean ± 95% confidence intervals of linear fit. Neuronal responses to active but not passive motion were significantly selectively suppressed in the combined condition. Black and gray stars represent example cells from (**a**). **d** Comparison of sensitivity of our Purkinje cell population to the active versus passive components of concurrent motion are shown, versus sensitivities to active and passive motion experienced alone ($n = 31$). Data were normalized to the sensitivity to the passive motion alone (i.e., left column). Significant differences were found using a two-sided paired-sample Student's $t$-test; $p$ values were adjusted for multiple comparisons using the Benjamini-Hochberg procedure (***$p < 0.001$). Data are presented as mean values ± SEM.

## Purkinje cells selectively respond to passive head motion that is experienced concurrently with active motion

Our findings above establish that the head motion response of Purkinje cells in the anterior vermis are markedly attenuated in the active condition. This then raises the question of whether the observed suppression is specific to the sensory consequences of active motion—consistent with the internal model hypothesis—or if instead the observed suppression is the result of a non-specific gain change that occurs during active self-motion. To distinguish between these two possibilities, we next recorded neuronal activity as monkeys generated voluntary head movements while simultaneously undergoing passive whole-body rotation. If the observed suppression were specific to the sensory consequences of active motion, then one would predict that a given neuron should (1) continue to robustly encode the component of the head motion caused by passively applied rotation, while (2) still

remaining relatively unresponsive to any actively generated motion of the head relative to space. Figure 3a illustrates the responses of two representative Purkinje cells during combined stimulation. Comparison of the firing rates with the head-velocity traces across the passive, active, and combined conditions reveals that, indeed, the example neurons selectively encoded the passive component of motion (passive head-motion-only prediction, blue trace) rather than absolute head-in-space motion (total head-motion prediction, dashed black trace).

The observations shown in Fig. 3a are summarized for the population in Fig. 3b–d. Figure 3b, c show a comparison of each Purkinje cell's sensitivity to the passive (Fig. 3b) and active (Fig. 3c) components of motion versus its sensitivity to passive motion applied alone. Overall, our population results show that Purkinje cells responded similarly to passive motion in both conditions (Fig. 3b); the slopes of

the regression lines were not different from unity ($p = 0.22$ and $0.14$ for bimodal and unimodal Purkinje cells, respectively). Thus, consistent with our first prediction above, neurons faithfully encoded passive head motion, even when it occurred concurrently with active head motion. Moreover, consistent with our second prediction, Purkinje cell responses to active head motion were also significantly attenuated in the combined condition for both unimodal and bimodal neurons (Fig. 3c $p = 0.01$ and $<0.001$, for bimodal and unimodal Purkinje cells, respectively). Figure 3d summarizes these observations for our population of Purkinje cells for head motion. Notably, neuronal response sensitivities to the passive and active components of head motion in the combined condition were comparable to those to passive versus active head motion alone. Hence, in accordance with the internal model hypothesis, our findings demonstrate that the reduction observed in the individual Purkinje cell responses is exclusive to the sensory consequences of active movement.

### The reduced sensitivity of Purkinje cells to active head motion can be accounted for by their sensitivity to neck motor commands

Above we have shown that anterior vermis Purkinje cells preferentially respond to passive stimulation. Neuronal responses are markedly reduced for comparable active versus passive head motion. The question thus arises: what accounts for the attenuation that is observed in response to active versus passive self-generated head movement? In response to passive head-on-body motion, the response of a given anterior vermis Purkinje cell was well predicted by the linear summation of its sensitivity to vestibular and neck proprioceptor stimulation (Supplementary Fig. 6; see also ref. 28). In the active condition, in addition to receiving sensory feedback, the brain also generates motor commands to activate the neck musculature. Accordingly, we hypothesized that the motor-related inputs to the anterior vermis might account for the reduced sensitivity of Purkinje cells to active versus passive head motion.

To investigate this hypothesis, we first tested whether Purkinje cells, in the absence of sensory stimulation, demonstrate modulation during the production of a motor command. Specifically, monkeys made active orienting movements as described above, but their heads were occasionally unexpectedly restrained on a subset of trials such that no actual head motion was accomplished (see "Methods"). To confirm that the monkeys generated intended but unrealized head movement commands, we measured the resultant neck torque. Figure 4a illustrates the firing rate of two representative Purkinje cells with respect to the torque generation onset (over a −200 to 200 ms timeframe), showing robust motor-related responses. Overall, we found that these two cells were representative of our population; the majority of Purkinje cells in our population (80%) responded during the production of a neck torque in at least one direction. Additionally, we observed heterogeneity across the population, with neurons demonstrating different response dynamics that either decreases or increases in activity relative to their resting rate (Fig. 4a, Purkinje cell 1 vs. 2, respectively). These results are summarized in the left and right panels of Fig. 4b for preferred vs. nonpreferred direction attempted movements, respectively. We considered the first 50 ms after active torque generation, to focus on responses before the activation of long-latency stretch reflexes or voluntary adjustments[29]. Generally, the firing rates of most Purkinje cells in our sample either increased or decreased in preferred direction after active torque generation, relative to their resting discharges (Fig. 4b, left), and increased for active torque generation in the nonpreferred direction (Fig. 4b, right). Further, as a population, Purkinje cell responses were significantly different from resting rate in the nonpreferred direction relative to passive head motions (Fig. 4b, left and right panels for preferred vs. nonpreferred; paired t-test, $p$ 0.69 vs. $p < 0.001$). Taken together, these results demonstrate most Purkinje cells encode neck motor signals

that could contribute to their differential coding of active versus passive self-motion.

Next, to further address our hypothesis, we explicitly assessed whether these motor-related responses could account for the reduction in Purkinje cells simple spike firing rate observed during active vs. passive head motion. To test this proposal, we first tested if we could predict a given neuron's sensitivity to active head-on-body motion (Fig. 5a, right column) based on the linear summation of its responses to comparable passive head motion (Fig. 5a, left column) and its response to neck motor commands alone measured during attempted head movement (Fig. 5a, middle column). As illustrated in Fig. 5a, a simple linear model well predicted the response of two representative Purkinje cell's response during active head motion. Figure 5b, c compare the gain and phase of the predicted response and observed responses to active head movements for each neuron in our population, in the direction with the largest response to the motor-related input (see "Methods"; comparison between the predicted and observed response are shown separately for each neuron in the direction with the largest response to the motor-related input, Supplementary Fig. 7). In general, our simple linear model well predicted the gain of a given Purkinje cell's response during active head motion ($R^2 = 0.74$, not significantly different than 1 slope, $p = 0.31$). Furthermore, the observed and predicted phases were also well matched ($R^2 = 0.88$, not significantly different than 1 slope, $p = 0.5$). These results thus provide evidence that the integration of motor and sensory information at the level of Purkinje cells underlies their differential encoding of active versus passive movements.

Lastly, to establish whether the mechanism underlying vestibular reafference suppression in Purkinje cells employs predictive motor-related signals, we quantified their response time course in this same condition. Figure 5d illustrates the average change in firing rate evoked during intended but unrealized active head-on-body movements, computed with values normalized across individual neurons (Fig. 5d, top; orange line, see Methods). Consistent with the existence of a predictive motor-related input, the onset of the average response reached statistical significance just before the attempted movement, occurring ~20 ms prior to initiation. We also employed a second method in which we computed the correlation between the observed firing rate during active head-on-body movements and that predicted based on its responses to comparable passive versus attempted movement (see "Methods"). Consistent with the results of our first method, this correlation reached significance just prior (~10 ms) to the onset of the movement (Fig. 5d, bottom, see Methods). Accordingly, the results of both analyses confirmed that Purkinje cells responses during intended but unrealized active movements occurred prior to movement - before there could have been any influence from sensory feedback pathways. Thus, taken together our results suggest that the mechanism underlying vestibular reafference suppression in Purkinje cells employs predictive motor-related signals.

### Linear combination of the heterogenous Purkinje cells accounts for the suppressed response in target neurons in the deep cerebellar nuclei

To summarize so far, we have shown that (i) the head motion responses of anterior vermis Purkinje cells to active movement are selectively attenuated relative to passive movement and that (ii) *individual* Purkinje cells integrate sensory and motor signals in a manner consistent with the observed suppression. At first glance, these findings are surprising as they show that an individual Purkinje cell is not able to provide the cancellation signal, given that Purkinje cells send inhibitory projections to the deep cerebellar and vestibular nuclei[25,26], which also show attenuated responses to active head motion (reviewed in ref. 30). Here, we provide an intuitive explanation of how a reduction in the inhibitory input from Purkinje cells can lead to the generation of the cancellation signal required to reduce the vestibular

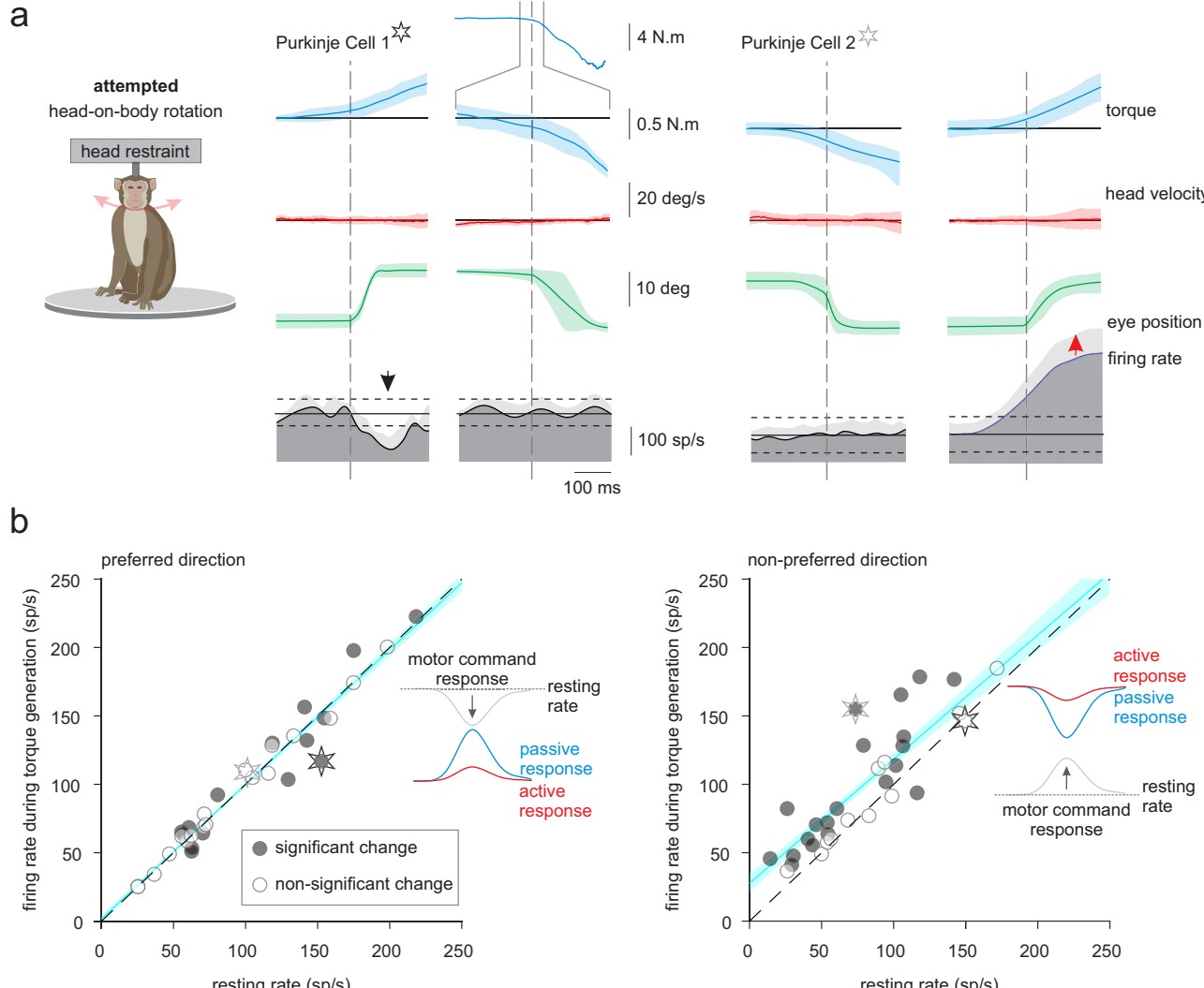

**Fig. 4 | Anterior vermis Purkinje cells respond to the production of a neck motor command signal made during attempted (but unrealized) orienting head movements. a** Two example Purkinje cells, one displaying excitatory and the other inhibitory modulation (black versus gray starts, respectively) during the generation of neck torques in the preferred (left) and nonpreferred (right) directions. The monkey's head was unexpectedly restrained as it oriented to an eccentric target, where the intended but unrealized movement was demonstrated by the production of neck torque. Traces are aligned on torque (blue) onset and show coincident head position (green), eye/gaze position (red). Each Purkinje cell's firing rates presented as trial-average values ± SD, and the superimposed dashed horizontal lines correspond to the 95% confidence intervals (CI) of resting discharge. **b** Population analysis: The mean firing rate of each Purkinje cell during periods of torque generation in the preferred (right) or nonpreferred (left) directions is plotted as a function of its resting rate. Significant torque-related changes in the firing rate for each cell and population of Purkinje cells were computed using two-sided paired-sample Student's *t*-test. For the population *p* values were adjusted for multiple comparisons using the Benjamini–Hochberg procedure (*p* = 0.70 and <0.001 for preferred and nonpreferred directions respectively). Stars represent the example Purkinje cells from (**a**). Blue lines and shading denote the mean ± 95% CI of linear fit (*n* = 34).

responses of these targeted neurons. First, consider a simple model in which only a *single* Purkinje cell projects to each deep cerebellar or vestibular nuclei neuron. If the Purkinje cell's response to head motion is attenuated in the active condition, then it is not possible to account for the suppression observed in the a given target neuron (Fig. 6a, Hypothesis 1A). Moreover, this would also be the case even if the sign of the response in the active condition is flipped (Fig. 6a, Hypothesis 1B). In contrast, consider a second model in which *multiple* Purkinje cells project to each deep cerebellar or vestibular nuclei neuron. If the distribution of the inputs from this convergent Purkinje cell population are attenuated but each neuron preserves its response direction during active compared to passive head-on-body movements (Fig. 6a, Hypothesis 2), then it is possible to completely cancel a given target neuron's response to passive stimulation. However, this is not a plausible approach, since the integration of inputs from an unrealistically

large population of Purkinje cells would be required to produce the required cancellation signal (i.e., >200 neurons; Supplementary Fig 8, see "Methods"). Finally, if the distribution of inputs from the convergent Purkinje cell population is not only attenuated but also spans both response directions during active as compared to passive motion (Fig. 6a, Hypothesis 3), then it is possible to generate the required suppression, with a realistically sized Purkinje cells population (i.e., ~40 neurons; Supplementary Fig. 8, see "Methods"). Accordingly, we hypothesized that the heterogeneous behavior of Purkinje cells, and in particular their tendency to reverse response direction under active conditions (i.e., the sensitivity of ~50% actually flip direction), is a key feature underlying the reafferent suppression that occurs in their target neurons within the deep cerebellar and vestibular nuclei.

To test our hypothesis directly on our experimental data, we first assessed the coding heterogeneity of our population of Purkinje cells

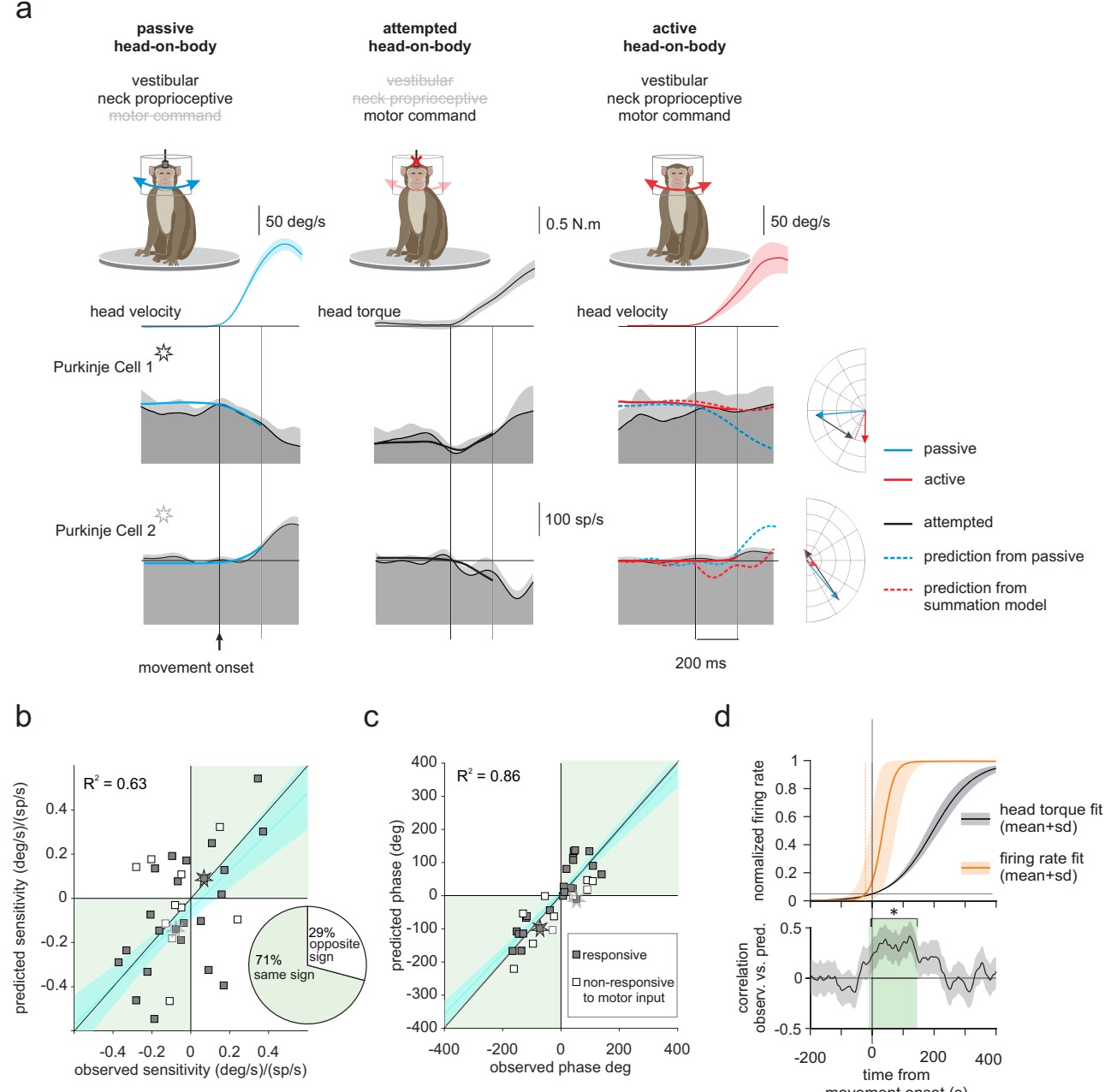

**Fig. 5 | The motor-related modulation of Purkinje cells can explain their reduced sensitivity to active versus passive self-motion. a** The firing rate of two example Purkinje cells during passive head motion (left, blue), attempted but unrealized head motion (middle, gray), and the active head movement (right, red). For the active head motion, predictions of the response - computed by adding the response during passive and attempted head movement conditions—are super-imposed. **b**, **c** Scatter plots comparing the observed vs. predicted gain and phase of each Purkinje cell's response in our population during active head movements. *Inset*: Percentage of the Purkinje cells with the same sign (green) and opposite sign (white) for the observed and predicted gain. In nearly all cases the prediction was

consistent with the observed response. Stars represent the example Purkinje cells from panel (**a**). Blue lines and shading denote means ± 95% CI of linear fits. **d** top: average responses computed from normalized fits to the firing rates of individual Purkinje cells (orange) during attempted head movements shown in comparison to the torque signal generated in this condition presented as trial-average values ± SD. bottom: The correlation between firing rate during active head-on-body movement and that predicted from the summation model. The gray shading denotes means ± 95% CI for the population of Purkinje cells ($n = 34$). Green shading denotes the time interval during which the correlation values are significantly higher than zero (two-sided permutation test, $p = 0.032$).

by plotting each neuron's active sensitivity as a function of its passive sensitivity (Fig. 6b, left). Most Purkinje cells (50%, yellow shaded region) displayed active attenuation that preserved the same response direction as in the passive condition (e.g., yellow stars in Fig. 6a). Additionally, a substantial percentage of the remaining Purkinje cells (40%, green shaded region) not only displayed active attenuation but actually reversed their response direction relative to the passive

condition (e.g., green stars in Fig. 6a). Overall, comparison across our neuronal population revealed considerable heterogeneity in the relationship between active and passive sensitivities. As reviewed above, anterior vermis Purkinje cells target neurons in rostral fastigial nucleus (rFN), the most medial of the deep cerebellar nuclei. Thus, for comparison, we plotted the active sensitivity as a function of its passive sensitivity for rFN neurons during the same conditions[31]. Notably, in

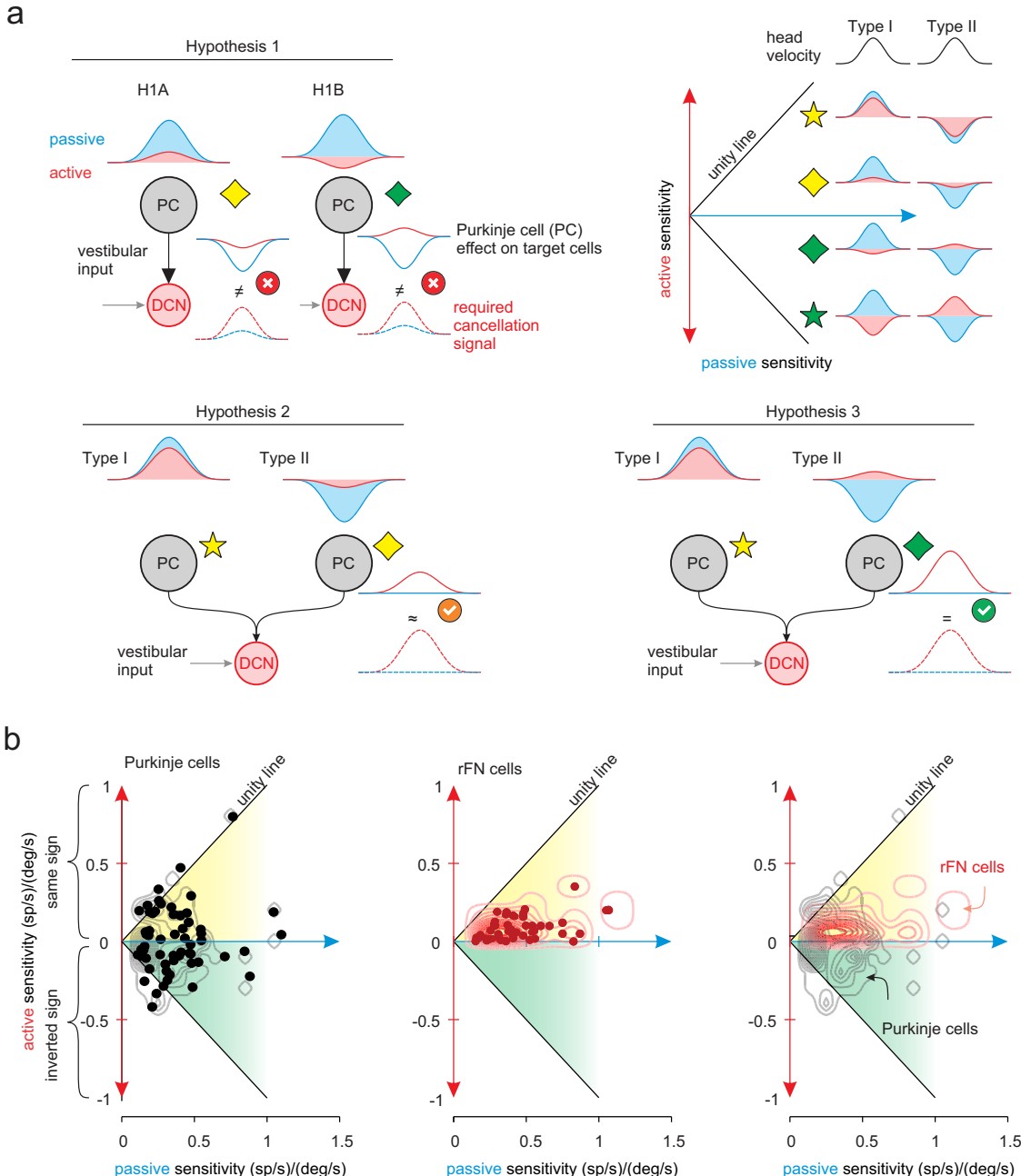

**Fig. 6 | Heterogeneity across the responses of individual Purkinje cells can account for the suppressed responses observed in target neurons in the deep cerebellar and vestibular nuclei. a** Three hypothetical Purkinje cell population responses. Only the combination of the Purkinje cells' responses with the response in the opposite directions could optimally generate the required suppression signal to their target neurons (Hypothesis 3). **b** Comparison of the population responses of the rostral fastigial (rFN) neurons (red) and Purkinje cells (black) during active vs. passive head-on-body movements ($n = 63$). The contour lines represent distributions of neural sensitivities.

contrast to the Purkinje cells of our present study, data for all rFN neurons (100%) are constrained to the yellow shaded region (Fig. 6b, center). The superposition of data from both populations (Fig. 6b, right) emphasizes that the attenuation of Purkinje cells is heterogeneous compared to their target neurons in rFN. Thus, taken together these findings are consistent with our hypothesis that, because Purkinje cell responses are not only attenuated but also spans both directions during active as compared to passive motion, they can generate the cancellation signal required to suppress vestibular reafference in their target neurons (Fig. 6a, Hypothesis 3).

Accordingly, we next tested whether this was the case. Specifically, to generate an inhibitory output that could (i) explain the suppression of vestibular input observed in target neurons during active motion (Fig. 7) and also (ii) eliminate the direct influence of motor commands on downstream targets, we used a simple linear model optimizing the weights of the activities of multiple Purkinje cells (Supplementary Fig. 9, see Methods). The latter condition is an important constraint, since target neurons in the vestibular and deep cerebellar nuclei - unlike Purkinje cells - are insensitive to motor commands[6,13]. Given that there was no explicit learning task in our

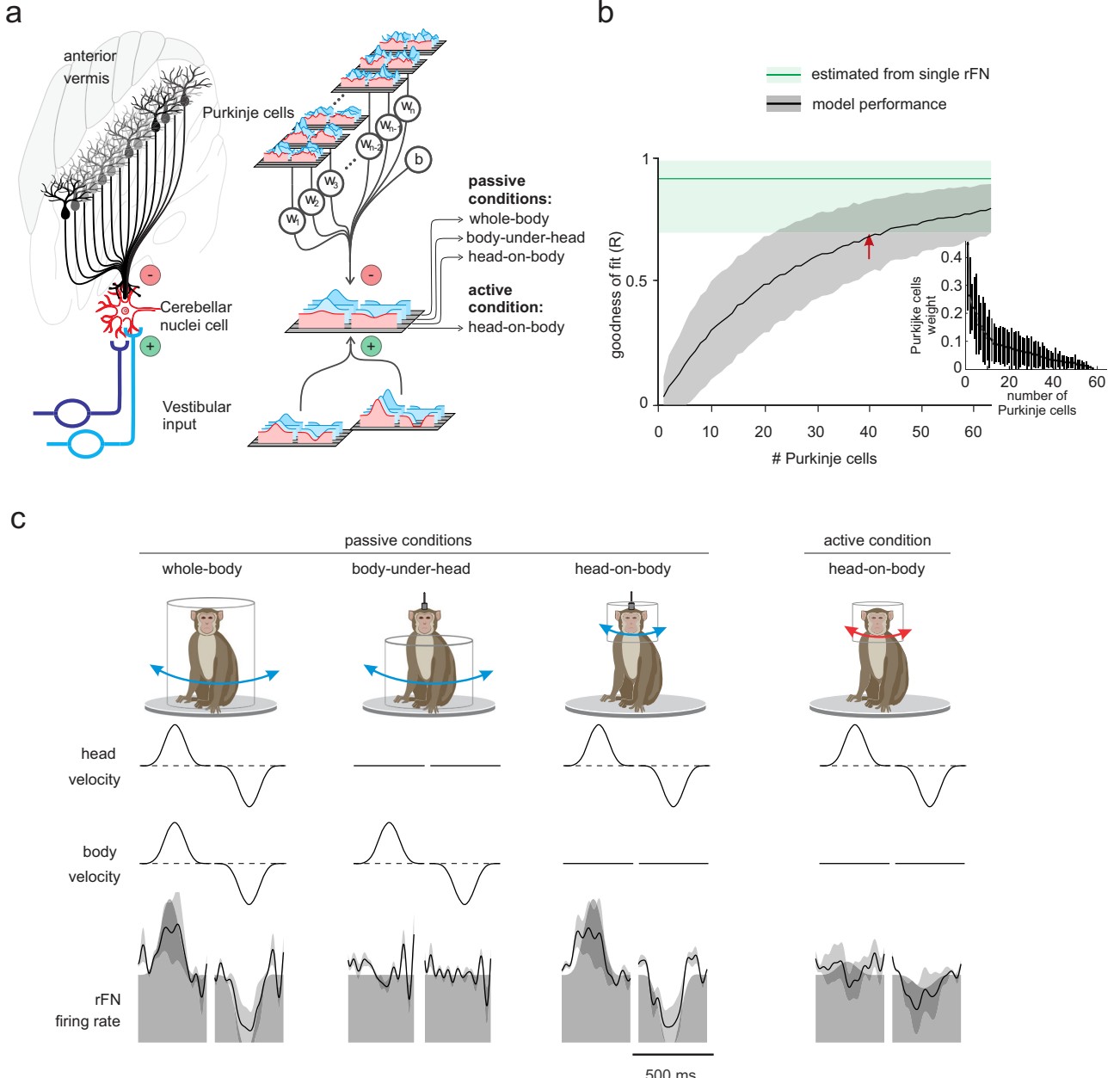

**Fig. 7 | A simple linear population model of Purkinje cell integration can explain the responses of unimodal target neurons in deep cerebellar nuclei across all self-motion conditions. a** Left: Illustration of the convergence of multiple Purkinje cells onto a single neuron in the rostral fastigial nucleus (rFN), with different shades representing theoretical differences in the weighing of each Purkinje cell's synapse with the target rFN neuron. Right: Schematic of the linear summation population model used to estimate the firing rate of a target neuron in the rFN. Each Purkinje cell's weight was optimized to generate the best estimate of the average bimodal rFN neuron across conditions[13]. **b** Model performance as a function of the number of Purkinje cells. The black curve corresponds to the model fit to the simple spike firing rates of all Purkinje cells in our population ($n = 63$), which were recorded during our three dynamic conditions (i.e., whole-body, body-under-head, and head-on-body passive movements and active head-on-body movements). The variability estimated from a population of rFN bimodal neurons previously described by refs. 13 is represented by the green shaded band. Data are presented as mean values ± 95% confidence interval. Inset: the distribution of computed weights for each Purkinje cell modeled during our three dynamic conditions with 40 Purkinje cells, sorted based on average weight. **c** Estimated model firing rates based on a population of 40 Purkinje cells superimposed on the actual average firing rate of a bimodal rostral fastigial nucleus (rFN) neuron (gray-shaded region). The firing rate estimations from models that included the four head/body rotation conditions are presented as mean values ± SD.

experiments, the weights in our modeling were held consistent across conditions, such that differences in neuronal responses across conditions arose from the differences that existed in the inputs that were available to the network (i.e., vestibular and proprioception signals, and/or motor commands). We then performed our modeling using two datasets. The first dataset included Purkinje cell responses recorded during our three passive (i.e., whole-body, body-under-head,

head-on-body rotations) and one active (head-on-body) condition. The second dataset included these same four conditions as well as our fifth condition in which head-restrained monkeys generated motor commands during intended but unrealized head motion (i.e., the attempted head movement condition). As expected, combining the activities of more Purkinje cells (i.e., increasing population size) led to an increase in the goodness of fit (Fig. 7b). In both cases, we found that,

the responses of either individual or a small group of Purkinje cells could not generate the required suppression signal. Instead, the combined activity of approximately 40 anterior vermis Purkinje cells produced responses that closely matched those previously reported for their target rFN and VN neurons. (Fig. 7b, red arrow). Moreover, using the second dataset we found that our model could negate the motor-related signals encoded by Purkinje cells in the attempted head movement condition. As a results, the target neurons do not respond to attempted head movements (refer to Supplementary Fig. 8, panel C, right) consistent with prior findings[6,13]. The inset in Fig. 7b shows the estimated model weights for our population of Purkinje cells and confirms that the model population-averaged response was not dominated by a small number of Purkinje cells. Thus, we conclude that a population of ~40 Purkinje cells can account for the reafference suppression of target neurons in the deep cerebellar and vestibular nuclei (Fig. 7c, Supplementary Fig. 9).

As noted above, Purkinje cells demonstrate considerable heterogeneity in their response dynamics to both vestibular and proprioceptive stimulation[28]. For example, Purkinje cells can i) show linear, V-shaped, or rectified tuning to vestibular input, ii) be either sensitive or insensitive to passive proprioceptive stimulation (i.e., bimodal vs. unimodal Purkinje cells), or iii) demonstrate excitatory versus inhibitory responses to ipsilaterally directed rotational vestibular stimulation (type I vs. type II responses). Thus, we next asked whether Purkinje cells with certain response attributes were weighted higher in our population model than others. However, we found that this was not the case. The model weight distributions were similar for linear versus V-shaped versus rectifying Purkinje cells, as well as for bimodal versus unimodal and type I versus type II Purkinje cells (Supplementary Figs. 11 and 12). Additionally, this was the case for Purkinje cells with and without motor-related responses (Supplementary Fig. 13). Thus, the model population-averaged response was not dominated by a small subclass of Purkinje cells.

Finally, in addition to their Purkinje cell inputs, the fastigial nucleus also receives mossy fiber input from the vestibular nuclei, reticular formation, and central cervical nucleus - areas that encode both vestibular and neck proprioceptive sensory information[32–35]. Accordingly, we tested whether accounting for this additional input altered our Purkinje cell population modeling results. Notably, the dynamics of these inputs have not been characterized during the head and/or neck rotations applied in the present study. Accordingly, we simulated mossy fiber input as a summation of vestibular and neck proprioceptive inputs where the gains and phases were randomly drawn from a distribution comparable to that previously reported in the vestibular nuclei[36], see Methods). We then further explored the effect of systematically altering this simulated mossy fiber input relative to the reference distribution of mossy fiber inputs by (i) doubling the gain, (ii) reducing the gain by half, (iii) doubling the phase, and iv) reducing the phase by half (see ref. 28). Overall, we found that the addition of such simulated mossy fiber inputs did not dramatically alter our estimate of the Purkinje cell population size required to suppress the vestibular reafference in rFN neurons during voluntary movements (~50 versus 40; Supplementary Fig. 13). Likewise, comparable results were obtained for model weight distributions as shown above (Supplementary Fig. 14).

## Discussion

Our results show that the marked suppression of self-motion responses in early vestibular pathways that are the consequence of active head movements can be explained by the responses of anterior vermis Purkinje cells. Purkinje cell simple spike activity was selectively and similarly suppressed for active head movements made in isolation and for active head movements in a condition where passive and active head motion was experienced simultaneously. Examination of Purkinje cells during attempted but unrealized head movements further

revealed that the majority of Purkinje cells encode neck motor-related signals that were consistent with their reduced sensitivity to active versus passive self-motion. Using a simple linear population model, we found that combining the inhibitory responses from ~40 Purkinje cells can account for the marked suppression that occurs in early central vestibular pathways. Thus, taken together, our findings reveal that individual anterior vermis Purkinje cells integrate motor with sensory signals consistent with the computation of an internal model that suppresses the sensory consequences of active self-motion required to ensure postural and perceptual stability.

### Purkinje cell responses provide a predictive suppression signal during active self-motion

Multiple lines of evidence in our present study indicate that the cerebellum plays an essential role in generating the predictive suppression signal that is required to distinguish the sensory consequences of actively generated self-motion (e.g. refs. 6–11,). First our experimental design, which included active head movements both made in isolation and made concurrently with simultaneous passive self-motion, eliminated the possibility that suppression is caused by a non-specific gain change to vestibular afferent input during active self-motion. Instead, suppression was both robust and selective for active head movements. Correspondingly, passive head movements were robustly and similarly encoded during passive head movements made in isolation as well as in this combined condition. Moreover, we found most Purkinje cells encoded motor-related signals in a manner consistent with their reduced sensitivity to active versus passive self-motion. Together, these findings suggest suppression of vestibular signals in the cerebellar anterior vermis during active self-motion is the result of motor-sensory predictions, rather than a non-specific suppression of vestibular afferent input.

The suppression of sensory input has been reported in other sensory systems during active behaviors, including in the primate somatosensory system[37–39], the crayfish mechanosensory system[40,41], and the cricket auditory system[42,43]. In these systems, however, there is evidence that presynaptic inhibition reduces the overall strength of the peripheral sensory signal transmitted to the central pathways. Thus, our present findings contrast with this prior body of work by showing that suppression in the mammalian vestibular system is selective to the component of sensory stimulation resulting from actively generated self-motion. Indeed, presynaptic inhibition at the level of the peripheral vestibular afferents would pose a major computational challenge to the brain during self-motion. On the one hand, the motor commands generated by the vestibulo-spinal reflex pathways are counterproductive when the behavioral goal is to actively move through space. Conversely however, the motor commands generated by the vestibulo-ocular reflex (VOR) are vital for effectively stabilizing gaze relative to space during our everyday activities. Indeed, consistent with the functional goals of these pathways, single-unit studies have established that efficacy of the VOR pathway is intact during active head turns when gaze is stable[44,45], while that of the vestibulo-spinal reflex pathways is markedly suppressed[6–11]. It is thus essential that vestibular afferents transmit a robust and accurate representation of head motion to central VOR pathways to ensure gaze stability regardless of whether head motion is passively or actively generated, as has been previously demonstrated[1–4]. In this context, our present results provide direct evidence that vestibular reafferent suppression is mediated centrally rather than peripherally. Specifically, we show that the output of the cerebellar anterior vermis is consistent with a forward model that selectively suppresses the reafferent component of the signal transmitted by vestibular afferents to the vestibular and deep cerebellar nuclei.

Movement-related modulation of the anterior vermis has been reported in studies of mice walking on a treadmill[46–49] as well as in freely moving conditions[50]. However, the origin of these signals and

whether they encode sensory feedback or predictive motor information to which sensory feedback could be compared, remains unknown. By recording the responses of Purkinje cells across a systematic series of passive stimulation and active behavioral conditions, we were able to explicitly dissociate sensitivities to passive vestibular and proprioceptive stimulation from motor-related responses. Furthermore, we found that most anterior vermis Purkinje cells demonstrated significant modulation when monkeys attempted head motion but were experimentally prevented from actually moving their head. Importantly, the presence of these motor-related signals in the anterior vermis contrasts with their complete absence at the level of their target neurons in both the vestibular and deep cerebellar nuclei[11,13]. Prior anatomical studies have established that cortical areas including primary motor cortex (M1) project to the anterior vermis[51]. Indeed, projections from motor cortex inputs contribute to the suppression of sound-evoked responses in the mouse auditory system during movement[52,53]. Additionally, movement signals could originate in subcortical structures, for example the lateral reticular nucleus[54], higher-order thalamus[55], and basal ganglia[56], since these areas encode movement as well as sensory information. Irrespective of the source of the motor signals, the presence of motor-related inputs in these Purkinje cells is consistent with a theorized role of the anterior vermis in generating an internal forward model of self-motion[16,57–59].

### The emergence of a predictive suppression signal through population coding

Our present findings further establish that individual Purkinje cells in the anterior vermis combine multiple streams of sensory information with motor information to compute the reafference cancellation signal required to distinguish between active and passive self-motion. Anterior vermis Purkinje cells send strong inhibitory inputs to the vestibular and deep cerebellar nuclei neurons[25,26], and these target neurons display markedly attenuated responses during active head movements (reviewed in ref. 30). Consequently, an unexpected result of this study is that the modulation of individual Purkinje cells decreases rather than increases during active movements. However, our results also reveal that Purkinje cells respond to motor-related signals as well as vestibular and neck proprioceptive sensory stimulation with substantial heterogeneity. Using a simple population model, we found that the number of Purkinje cells required to account for the suppression observed in their target neurons (i.e., 40–50 Purkinje cells) matched the experimentally determined convergence ratio of the Purkinje cells onto these target neurons[60,61]. Given that there is such response heterogeneity across individual neurons, their responses effectively comprise a form of "expansion coding", which theoretically provides a temporal basis set (e.g.[62–64,]) for generating the precise predictive suppression signal required to distinguish the sensory consequences of actively generated self-motion in downstream pathways.

Interesting, our finding that the firing rates of deep cerebellar nuclei neurons can be reconstructed as a weighted linear sum of cerebellar Purkinje cell (and Mossy fiber) inputs aligns with Tanaka et al.'s research[65,66] on wrist movements. In their study, they were also able to predict the responses of Mossy fibers based on the activity of deep cerebellar nuclei from the previous trial, in a manner similar to a Kalman filter. Given that the suppression of vestibular reafference in our system represents direct evidence of a computation reliant on the internal forward model, it would be intriguing to explore whether a similar mechanism functions within this system. This could provide a promising avenue for future research into forward models in the cerebellum. It is further noteworthy that the weights in our modeling were held consistent across conditions. As a result, the differences observed in Purkinje cell firing across conditions were due to the differences in the inputs available to the network, specifically vestibular, proprioception, and/or motor-related inputs. Understanding whether and

how these synaptic weights change during a learning task (i.e. ref. 67,) is interesting direction for future work.

Taken together our findings thus raise the question: why does the brain use this complex approach in favor of a simpler approach? For example, one might posit that enhancing the responses of a relatively homogenous population of Purkinje cells during active movements would be a more straightforward computational strategy. We propose that the observed suppression combined with coding heterogeneity serves two key functions. First, the encoding of sensorimotor inputs is advantageous in the context of motor learning, since it provides redundancy in the synaptic weights that generate a new desired outcome (i.e., learning in weight space, reviewed in ref. 68). When adaptation to a new environment is required, fast and reliable learning can be achieved by small changes distributed across individual Purkinje cells. Second, while the enhancement rather than inhibition of the Purkinje cell modulation may provide a more intuitive solution, it has disadvantages in the context of optimal information processing. Notably, enhancement of a Purkinje cell's modulation transmits information less efficiently than inhibition since it requires the generation of more action potentials, thereby resulting in a higher metabolic energy cost[69–72]. Finally, we note that Purkinje cells generate complex spikes due to inputs from olivary climbing fibers and it has been reported that pooling the responses of Purkinje neurons in the oculomotor vermis based on the directionality of their complex spike sensitivity can improve population-based predictions (e.g. ref. 73,). To date, however, no studies have examined complex spikes in the vestibular region of the anterior vermis in general or more specifically whether they correspond with patterns of Purkinje cells convergence onto rFN. Future studies are required to test whether pooling based on complex spikes attributes can further improve modeling predictions.

### Implications for ensuring postural and perceptual stability

The ability to estimate unexpected self-motion to maintain postural and perceptual stability requires the integration of motor-related and sensory signals. In this context, our current findings have important functional implications. First, they provide direct insight into the mechanism underlying prediction-based suppression in vestibulo-spinal pathways during active movements. The efficacy of vestibulo-spinal pathways, quantified by measurement of the vestibular sensitivities of vestibular and deep cerebellar nuclei to self-motion, is markedly suppressed during active head movements[11,13]. Prior to our study, the source of the signal required to cancel the intact peripheral vestibular afferent input[1–4] to vestibulo-spinal pathways was unknown. Additionally, the efficacy of the posterior thalamocortical vestibular pathway is suppressed during active head movements[12]. Given that this pathway receives input from the same class of neurons in the vestibular nuclei that demonstrate reafferent suppression[74], it is likely that this cerebellum-based computation correspondingly contributes to ensuring perceptual as well as postural stability during self-motion (reviewed in ref. 75).

## Methods

### Experimental model and subject details

Animal experimentation: All experimental protocols were approved by the Johns Hopkins University Animal Care and Use Committee and were in compliance with the guidelines of the United States National Institutes of Health (PR19M408). The cerebellar recordings were conducted in two male macaque monkeys (Macaca mulatta). The animals were housed on a 12-h. light/dark cycle. The recording sessions were about three times a week, for approximately 2 h. each session. Both animals had participated in previous studies in our laboratory, were in good health, and did not require any medication.

## Surgical procedures

The two animals were prepared for chronic extracellular recording using aseptic surgical techniques. Animals were pre-anesthetized with ketamine hydrochloride (15 mg/kg i.m.) and injected with buprenorphine (0.01 mg/kg i.m.) and diazepam (1 mg/kg i.m.) to provide analgesia and muscle relaxation, respectively. Loading doses of dexamethasone (1 mg/kg i.m.) and cefazolin (50 mg/kg i.v.) were administered to minimize swelling and prevent infection, respectively. Anticholinergic glycopyrrolate (0.005 mg/kg i.m.) was also preoperatively injected to stabilize heart rate and to reduce salivation, and then again, every 2.5–3 h. during surgery. During surgery, anesthesia was maintained using isoflurane gas (0.8–1.5%), combined with a minimum 3 l/min (dose adjusted to effect) of 100% oxygen. Heart rate, blood pressure, respiration, and body temperature were monitored throughout the procedure. During the surgical procedure, titanium post for head immobilization and recording chambers were fastened to each animal's skull with titanium screws and dental acrylic. Craniotomy was performed within the recording chamber to allow electrode access to the cerebellar cortex. An 18-mm-diameter eye coil (three loops of Teflon-coated stainless-steel wire) was implanted in one eye behind the conjunctiva. Following surgery, we continued dexamethasone (0.5 mg/kg i.m.; for 4 days), anafen (2 mg/kg day 1, 1 mg/kg on subsequent days), and buprenorphine (0.01 mg/kg i.m.; every 12 h. for 2–5 days, depending on the animal's pain level). In addition, cefazolin (25 mg/kg) was injected twice daily for 10 days. Animals recovered in 2 weeks before any experimenting began.

## Data acquisition

During the experiments, the monkey sat in a primate chair secured to a turntable, and their head was centered in a coil system (CNC Engineering). Extracellular single-unit activity was recorded using enamel-insulated tungsten microelectrodes (Frederick-Haer). The location of the anterior vermis of the cerebellar cortex was determined relative to the abducens nucleus identified based on stereotypical neuronal responses during eye movements, and Purkinje cells were identified based on their characteristic complex spike activity. The angular velocity of the turntable was measured using a gyroscope sensor (Watson Industries, Eau Claire, WI). The monkeys' gaze and head angular positions were measured using the magnetic search coil technique. The neck torque produced by the monkey against its head restraint was measured using a reaction torque transducer (QWFK-8M; Honeywell, Canton, MA). All analog behavioral signals were low-pass filtered with a 125 Hz cut-off frequency and acquired at 1 kHz. The neural activity was recorded at 30 kHz using a data acquisition system (Blackrock Microsystems). Action potentials from the neural recording were sorted using a custom MATLAB GUI (MathWorks), which provides threshold, clustering, and manual selection/removal methods.

## Active and passive self-motion paradigms

Two monkeys were trained to orient to a target projected onto a cylindrical screen located 60 cm away from the monkey's head. Each neuron's insensitivity to saccades and ocular fixation was confirmed by having the head-restrained monkey attend to a target that stepped between horizontal positions over a range of ±30°. To ensure consistency, we only included cells that exhibited at least 10 head movements in each direction. Each neuron's lack of response to eye movements was further confirmed by absent responses to smooth pursuit eye movements during sinusoidal target motion (0.5 Hz, 40°/s peak velocity). Histological analysis confirmed that the Purkinje cells were located in lobules II–V of the anterior vermis, ~0–2 mm from the midline. Further, while we first tested the

vestibular sensitivity of individual neurons, we did also test whether neurons that were insensitive to vestibular stimulation responded to neck proprioceptive stimulation. Consistent with Manzoni and colleagues' prior studies in anesthetized cat (12%)[76], we found that only a small portion of Purkinje cells (~10%) fell into this latter category.

Passive vestibular (i.e., whole-body rotation) and proprioceptive (body-under-head rotation) stimuli were first applied as described by Zobeiri and Cullen[28]. Stimuli were characterized by "active-like motion" trajectories corresponding to those produced during active head-unrestrained gaze shifts (see "head-free paradigms" below). In addition, neural sensitivities to both proprioceptive and vestibular stimulation were assessed by passively rotating the monkey's head relative to its stationary body (i.e., head-on-body rotations) with this same trajectory.

After a neuron was fully characterized in the head-restrained condition, the neuron's response was then recorded as the monkey's head was carefully released to maintain neuronal isolation. Once released, the monkey was able to rotate its head freely in the yaw axis. Neurons were recorded as monkeys made ±30° active head movements for a juice reward while their body was (1) stationary, and (2) simultaneously passively rotated (1 Hz, 40°/s peak velocity). The latter paradigm allowed us to characterize the response of the Purkinje cells to concurrent voluntary and passive movements and was termed the 'combined' condition. Finally, neuronal responses were recorded during an 'attempted head movement' condition. In this condition, we applied random brakes to the head movements after the monkey had oriented to a target for >500 ms, just prior (100 ms) to presenting the next target. To do this, we activated an electromagnetic clutch (Placid Industries), attached to the head-holder. Such breaks were applied unexpectedly, in head-unrestrained monkeys, for a small subset of trials (less than 5%). We then measured the torque while the monkey, without knowledge of the imposed restraint, tried to make head movements.

The large torques measured during this paradigm (>1 Nm) verified that monkeys generated motor commands that were unrealized due to the restraint, thereby allowing us to determine the effect of the motor-related signals.

## Data analysis

**Analysis of neuronal discharge dynamics.** Data were imported into the MATLAB (MathWorks) programming environment for analysis, filtering, and processing. Neuronal firing rate was computed by filtering spike trains with a Kaiser window at twice the frequency range of the stimulus[77]. We first verified that each neuron neither pauses nor bursts during saccades and was unresponsive to changes in eye position during fixation. We then used a least-squares regression analysis to describe each Purkinje cell's simple spike response to whole-body and body-under-head rotations:

$$\hat{fr}(t) = b + c_{p,i}X_i(t) + c_{v,i}\dot{X}_i(t) + c_{a,i}\ddot{X}_i(t) \tag{1}$$

where $\hat{fr}(t)$ is the estimated firing rate, b is a bias term, $c_{p,i}$, $c_{v,i}$, and $c_{a,i}$ are coefficients representing the position, velocity, and acceleration sensitivities respectively to head ($i = 1$) or body motion ($i = 2$), and $X_i$, $\dot{X}_i$ and $\ddot{X}_i$ are head ($i = 1$) or body ($i = 2$) position, velocity and acceleration (during whole-body and body-under-head rotations), respectively. This least-squares regression was solved for non-negative and non-positive criterion to ensure sign consistency across estimated coefficients. For each model coefficient in the analysis, we computed 95% confidence intervals using a nonparametric bootstrap approach ($n = 2000$)[78,79]. All non-significant coefficients were set to zero. We then used coefficients

to estimate the sensitivity and phase of the response using the following equations:

$$Sensitivity = \text{sgn}\left(c_{a,i}, c_{v,i}, c_{p,i}\right) \times \sqrt{\frac{\left((2\pi f)^2 c_{a,i} - c_{p,i}\right)^2 + \left(2\pi f c_{v,i}\right)^2}{(2\pi f)^2}} \quad (2)$$

$$Phase = tan^{-1}\left(\frac{(2\pi f)^2 c_{a,i} - c_{p,i}}{2\pi f c_{v,i}}\right) \quad (3)$$

For which $f = 1 Hz$ to match the duration of half-cycle of movements (500 ms) and the sign term (i.e., $\text{sgn}(c_{a,i}, c_{v,i}, c_{p,i})$) equals either 1 or −1 for positive versus negative coefficients, respectively. The sensitivity of the Purkinje cells to the neck proprioceptive stimulation (during body-under-head rotations) was used to categorize the cells into unimodal (zero sensitivity) and bimodal (non-zero sensitivity).

We used a similar approach to estimate sensitivities to passive and active head-on-body movements. Since in these conditions, it is not possible to dissociate neck proprioceptive and vestibular sensitivities, we estimated them as a single coefficient. To quantify the ability of the linear regression analysis to model neuronal discharges, the variance-accounted-for (VAF) for each regression equation was determined by subtracting the residual variance from the total variance and divide by the total variance. Values are expressed as mean ± SD and two-sided paired-sample Student's $t$-tests were used to assess differences between conditions.

Neuronal tuning of the responses to active and passive movements was further categorized as linear, rectifying, or V-shaped. Linear neurons demonstrated increased and decreased firing rates in the preferred and nonpreferred directions, respectively. The difference between the magnitude of sensitivities in each of the two directions was within 0.2 (sp/s)/(°/s). Rectifying neurons demonstrated increased firing rate in the preferred direction and minimal modulation (i.e., sensitivity smaller than 0.2 (sp/s)/(°/s)) in the nonpreferred direction. V-shaped neurons demonstrated an increased firing rate in both directions. The difference between the magnitude of their sensitivities in each of the two directions was within 0.2 (sp/s)/(°/s). Finally, neurons that did not fit any of these criteria were characterized as 'other'. Note that V-shaped neurons were categorized as Type I or II based on the direction for which their vestibular sensitivity was larger since the magnitudes of their responses in each direction were not identical.

To quantify the encoding of vestibular signals during concurrent active and passive movement, we computed neuronal sensitivities to active head-on-body versus passive whole-body rotation in the combined condition. We then normalized these sensitivities by the sensitivity to passive sinusoidal whole-body rotation alone to compute reafference and exafference ratios. To quantify the encoding of motor commands, we analyzed neuronal responses during the 'attempted head movement' condition. We identified neurons responding to the condition by calculating the change in firing rate between 100–300 ms before and 0–100 ms after each attempted movement. A two-sided permutation test revealed neurons with significant torque-related changes, which we classified as motor-related cells.

Finally, to estimate neuronal response timing relative to torque generation in the attempted but unrealized head movement condition, we used two Methods. First, we fit sigmoid functions to both torque signals and firing rate data during attempted head movement and then utilized the temporal parameters obtained from the best-fit sigmoid function to calculate the onset of movement and the associated firing rate change. These values were defined as the point at which each of the sigmoid fits reached 5% of its range. Second, we assessed the time at which the correlation between i) neural firing during active head movements and ii) that predicted-based responses to comparable passive versus attempted movements reached significance, using a 40 ms

sliding window with 1 ms overlap. The significance level was determined based on the 95% confidence intervals of the correlation observed during a baseline period (i.e., 100–300 ms before movement onset).

**Test of linear summation model.** To test whether the summation of motor and sensory (i.e., motor command, vestibular, and neck proprioception) responses could explain the attenuated response of the Purkinje cells to voluntary head movements we tested a simple linear model. Purkinje cell responses in three conditions: i) passive head-on-body, ii) attempted head movement, and iii) active head movement were described using the dynamic representation described above (i.e., Eq. 1). For the attempted head movement condition, we used average head-velocity generated during active head-on-body condition as an estimate of intended head-velocity. In this approach, we first selected the firing rate from 50 ms before to 100 ms after the onset of head movement/attempted movement. Then, we fitted the unitless dynamic model to compute the gain and the phase of the vector representation of the firing rate during each of the three conditions. Then the vector summation of the responses to the "passive head-on-body" and "attempted head movement" conditions were used to predict the vector representation (i.e., gain and phase) of the firing rate during "active head movement" condition.

**Simulation of population cancellation signal.** In order to investigate how different hypothetical populations of Purkinje cells could generate a cancellation signal to suppress the vestibular responses of target rFN neurons during active head movements, we employed the following linear model:

$$FR_{cancellation} = \sum_{i=1}^{N} w_i \times Pcell_i \quad (4)$$

Here, $FR_{cancellation}$ represents the opposite value of vestibular afferent inputs during active head movements and is zero during passive condition. $Pcell_i$ represent firing rate of Purkinje cells for the entire movements, for two directions, across all conditions. The weights ($w_i$) correspond to the connections from Purkinje cells to rFN neurons, all of which are considered non-positive corresponding to their inhibitory effect. To test Hypothesis 2 from Fig. 6, we exclusively considered Purkinje cells with similar sensitivity signs in both active and passive conditions. For Hypothesis 3, half of the selected Purkinje cells showed the same sensitivity sign, while the other half displayed opposite signs in two conditions. To examine the impact of an increased number of Purkinje cells in the model beyond our dataset, we augmented the existing cells using a generative model. This generative model produced firing rates for the additional Purkinje cells, while ensuring similar dynamics (including sensitivity and phase distribution) as the Purkinje cells in our dataset.

**Population modeling of Purkinje cells.** To determine whether integrating the activities of multiple Purkinje cells could explain the response of their target neurons in the rFN, we used the linear model below:

$$\hat{rFN} = \sum_{i=1}^{N} w_i \times Pcell_i \quad (5)$$

where $\hat{rFN}$ is a reconstructed firing rate response of an rFN neuron. The $w_i$ corresponds to weights of connection from Purkinje cells to an rFN neuron, which all considered non-positive to reflect inhibitory synapses from Purkinje cells to rFN neurons. $Pcell_i$ is the observed simple spike firing rate of $N$ Purkinje cells, where $N$ is a number between one and the total number of Purkinje cells in the dataset. We considered the dynamics of firing rates in relation to the movements for the entire movement for two directions across all conditions.

For each $N$ we used a bootstrapping approach to find the 95% confidence intervals of the goodness of fit ($R^2$) as well as the model predictions.

To model the population response of Purkinje cells during the 'attempted head movement' condition, we first fit a Gaussian function on coefficients that reproduced the dynamics of the responses of the 34 Purkinje cells that were recorded during this condition. Next, we used the parameters of these Gaussian functions to find a normal distribution representing the responses of Purkinje cells. Then, for the remaining Purkinje cells that were not recorded during the 'attempted head movement' condition, we augmented synthesized responses by drawing from this normal distribution.

Finally, we modeled the contribution of the mossy fiber input to the rFN as a summation of independent responses to vestibular and neck proprioceptive stimulation. To simulate the mossy fiber input, we randomly selected response gains and phases from normal distributions that described the responses of neurons in the vestibular nuclei (i.e., $0.6 \pm 0.1$ (sp/s)/(°/s) and $20 \pm 5°$, respectively), and repeated this for a total of 1000 simulations. We further assessed the robustness of our modeling performance by modifying these distributions. Specifically, we tested the effect of either doubling or halving the gain (i.e., $1.2 \pm 0.2$ and $0.3 \pm 0.05$ (sp/s)/(°/s), respectively) or phase (i.e., $40 \pm 10°$ and $10 \pm 2.5°$, respectively), to produce specific simulations based on each of the resulting four modifications of the original distribution.

### Statistics and reproducibility

We employed a two-sided paired-sample Student's $t$-tests to assess the statistical significance of differences between groups. To control for multiple comparisons, we applied the Benjamini-Hochberg (BH) procedure, with a significance level of 0.05. No statistical method was used to predetermine sample size. The neurons that were not responding to the vestibular stimulation were excluded from the analyses. The experiments were not randomized. The Investigators were not blinded to allocation during experiments and outcome assessment.

### Reporting summary

Further information on research design is available in the Nature Portfolio Reporting Summary linked to this article.

## Data availability

The processed data for all figures in this manuscript are available through the corresponding Source Data file. The raw dataset on which they are based is available from the corresponding authors upon request. Source data are provided with this paper.

## Code availability

The custom written codes for this manuscript have been deposited on GitHub (https://github.com/omidzobeiri/zobeiri_cullen_2024.git).

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

## Acknowledgements

This research was supported by grants R01-DC002390 and R01-DC018061 from the National Institutes of Health (KEC). We would like to thank Dale Roberts for his technical support and Drs. Robyn Mildren, Pum Wiboonsaksakul, Lex Gómez, and members of the Cullen lab for helpful discussions.

## Author contributions

O.Z. and K.C. conceived the study and designed experiments; OZ. performed experiments and analyzed data. O.Z. and K.C. discussed the data and wrote the manuscript.

## Competing interests

The authors declare no competing interests.
