## [Peer Review File · Nature Communications]

REVIEWER COMMENTS

Reviewer #1 (Remarks to the Author):

General comments

In the last sentence of the Introduction, the authors stated “Thus, taken together our results constitute the first direct demonstration of the widely held view that the cerebellum integrates movement-based predictions with actual sensory feedback to cancel the sensory consequences of active self-motion.” Unfortunately, I am not yet convinced as summarized below.

Specific comments

Major:

1. The presentation of this manuscript appears to depend too heavily on the two examples of Purkinje cells (PCs) (Cell 1 and Cell 2) in Figs. 1, 3, 4 and 5. If you used different neurons for different figures, you should mention it in the figure legends.

2. The authors modeled PC activities as a linear sum of activities proportional to the position, the velocity, and the acceleration (page 18, Equation 1). Nevertheless, almost the entire analysis in this manuscript is somehow dedicated to analyses of velocity-related PC activities. Internal models are not dedicated to the prediction of velocity.

3. The authors raised important points in the Introduction as follows -(1)(2)(3):

(1) Page 3, lines 14-17: “The vestibular afferents of the VIII nerve encode sensory signals in a manner that is not dependent on context; their responses are comparable whether self-motion is unexpected or intentional”.

(2) Page 3, lines 17-21: “In contrast, the vestibulospinal pathway neurons within the vestibular nuclei, which are directly targeted by these same afferents, selectively respond to vestibular sensory stimulation during unexpected (i.e., passive) self-motion, as do neurons in the ascending posterior thalamocortical vestibular pathway. This selective encoding of unexpected vestibular information provides a neural mechanism for ensuring postural and perceptual stability during self-motion”.

(3) Page 3, lines 24-28: “To date, however, the source of the suppression signal that cancels peripheral vestibular input to central pathways during expected (i.e., active) self-motion remains unknown. Importantly, vestibular nuclei neurons are capable of distinguishing between expected and unexpected vestibular stimuli, even when these stimuli occur simultaneously”.

Based on (1), (2), (3), and because the vestibular nuclei receive major inputs from the VIII nerve and PCs in the cerebellar cortex, it is most likely that the suppression signal comes from the cerebellar cortex.

Namely, the PCs send it to the vestibular neurons. This inference predicts that PC activities encode the motor command's sensory consequences, rather than simple attenuation of sensory responses as demonstrated in Figs. 1, 2, and 3. This inference also predicts that PC activities represent the negative of the canceling signal because PCs are inhibitory. Nevertheless, neither prediction does not appear to match the results of this manuscript.

4. Page 2, lines 9-10, Abstract: "Purkinje cells exhibit selective suppression for actively versus passively generated vestibular input, even when passive stimulation is experienced simultaneously.": This result appears to suggest that the source of the suppression signals mentioned above are generated before PCs. Then where? This issue should be explained and discussed.

5. PCs are GABAergic and exert tonic inhibition on vestibular nuclei neurons. Therefore, the suppression or decrease of "actively generated vestibular" response in PCs should result in disinhibition (i.e., facilitation) of target vestibular nuclei neurons as demonstrated before (Ishikawa et al., 2014, PLoS ONE, 9(10): e108774). It appears that this paper has not factored in this basis of cerebellar physiology.

6. Page 7, lines 27-30: "In the active condition, in addition to receiving sensory feedback, the brain also generates motor commands to activate the neck musculature. Accordingly, we hypothesized that the motor-related inputs to the anterior vermis might account for the reduced sensitivity of Purkinje cells to active versus passive head motion". I agree that this is a possibility. Then where can we find the evidence of the efference copy activities of PCs in this manuscript?

7. Fig. 4, the legend: "The monkey's head was unexpectedly restrained as it oriented to an eccentric target, where the intended but unrealized movement was demonstrated by the production of neck torque".

I cannot agree to assume the neck torque (blue lines in Fig. 4) as "attempted" because the gradually incremental torque profile (Fig. 4) must be different from a torque profile for a natural neck movement without a restraint (e.g., Fig. 1). I would recommend the term "perturbed neck torque". Note that the temporal patterns of PC activities (gray areas) do not match with those of neck torque (blue lines).

8. The authors may want to compare the present study with a recent study that has provided neural evidence of the cerebellar forward model in the cerebrocerebellum (Tanaka et al. (1), (2)). The cerebrocerebellum, forming a closed loop between cortical motor areas, receives both the efference copy and somatosensory inputs required for a neuronal substrate to serve as a forward model. They recorded neural activities in monkeys performing step-tracking movements. The activities of mossy fibers (MFs) at time $t+1$ (the neural activity encoding a future state) were well reconstructed as a weighted sum of dentate nucleus cells (DNCs) at time t (the neural activity representing the current output of the cerebellum). This analysis provided the first direct evidence for the internal forward model hypothesis of

the cerebrocerebellum. In addition, they established a set of linear equations to relate the activities of the cerebellar neurons. First, the PC activities were reconstructed linearly as the weighted sum of the MF activities. Second, the DNC activities were reconstructed linearly as the weighted sum of the PC activities and the MF activities. These linear equations were compatible with those of the optimal predictor known as the Kalman filter (Tanaka et al. (1), (2)). In the Kalman-filter framework, the predictive step computes the predictive state (PC activities) from the current state (MF activities), and the filtering step (DNC activities) integrates the predictive state (PC activities) and sensory feedback from the periphery (another set of MF activities).

(1) Tanaka et al. (2019) Neural evidence of the cerebellum as a state predictor. *Cerebellum*. 18, 349-371.

(2) Tanaka et al. (2020) The cerebro-cerebellum as a locus of forward model: a review. *Front Sys Neurosci*. 14:19.

9. In Fig. 3, the phase lags (i.e., delays) between the passive head velocity and PC activities appear significantly different for PC1 and PC2. I would expect consistent delays for reliable prediction. How do you explain the variation?

Specific comments

Minor:

1. There is little description of how the example PCs (in Figs. 1, 3, 4, and 5) are representative of the whole population of PCs recorded in this study. It is helpful to identify these representative neurons in plots of Figs 2ABC, 3BC, 4B, and 5BC.
2. The location of the PCs in the cerebellar cortex is essential information to be presented as an inset.
3. For Figs. 3, 4, and 5, it is necessary to describe calculated values for the example neurons.
4. In Fig. 4A, PC1, the onset of suppression (left) appears too late for the torque change. In addition, the correlation of their temporal patterns is poor.
5. In Fig. 4, where is the torque onset? The hatched lines in the center? However, the torque level is already significantly deviated from the baseline at the center.

Reviewer #2 (Remarks to the Author):

The ms by Zobeiri and Cullen is the latest in a series investigating vestibular coding during active and passive self-motion. Elegant prior work by the same group has shown that neurons in the rostral fastigial nucleus (rFN) of the cerebellum (Brooks and Cullen, 2013, 2015), as well as neurons in the vestibular nucleus, selectively encode passive head rotation while remaining insensitive to vestibular input when it is caused by the animal's own movement. The hypothesis from prior work is that motor-based predictions cancel the self-generated input, however, the important questions of where/how the cancellation signal is constructed remain unclear. The present study extends the investigation from the output stage of the cerebellum (rFN neurons) to Purkinje cells of the anterior vermis that are presumed to inhibit rFN cells. Response properties to active and passive motion regimes tested previously in rFN and vestibular nucleus are tested in Purkinje cells. A model of rFN responses is proposed in which cancellation of vestibular input during active head rotation can be accounted for based on a weighted sum of ~40 Purkinje cell inputs.

This is a logical and important next step in the authors' investigations with broad potential relevance to understanding cerebellar function and sensorimotor processing. However, I have several major concerns. First, the connection to the closely related past work is poorly communicated, making it difficult to appreciate the new results being shown. The abstract states "Here, we provide the first direct evidence that the cerebellum combines incoming sensory signals with internally generated predictive motor signals to selectively cancel actively generated sensory input." I was confused by this, because the authors have already demonstrated adaptive cancellation of self-generated input in the cerebellar nucleus in, at least, two prior papers. The intro focuses on vestibular nucleus and barely mentions rFN or the fact that the anterior vermis PCs recorded here presumably inhibit rFN (but not VN presumably). Also, what about connections between rFN and VN? Although the significance here lies in understanding how the cancellation in rFN is done (rather than in understanding PC responses in isolation), this is not communicated clearly enough until the final section of the results. Does the above sentence from the abstract refer specifically to the finding that motor signals in Purkinje cells appear to partially cancel self-generated input? If so, this raises the question of whether this "computation" at the level of the PCs actually supports the cancellation at the output stage. This is far from clear. Since PCs are inhibitory, they would be expected to transmit the motor signals to rFN cells instead of cancelling them out (as the authors point out). With appropriate weighting, the passive responses of different Purkinje cells can be averaged out, allowing the residual responses to self-generated motion to serve as the cancellation signal. However, in this scheme it is unclear why cancellation at the level of PCs needs to happen in the first place. Framing what happens in individual PCs as an elegant computation when it needs to be immediately undone by pooling populations at the level of the nucleus is confusing and distracts from the actual biological findings which are valuable even though they may be nuanced/complex.

Second, the modeling and related discussion can be expanded upon and improved. The authors should attempt to fit other aspects of rFN responses that they have previously recorded in Brooks and Cullen 2013, 2015. For example, can the same weighted population of PCs explain the prior finding that rFN neurons don't respond to attempted head movements? I believe the authors have shown previously that cancellation does not occur if the motor command does not match the proprioceptive input. Can the model explain this? Prior work in oculomotor vermis has shown that summing PC responses pooled

based on complex spike sensitivity input leads to useful signals (work from Herzfeld and Shadmehr should be cited). This makes sense because PCs that share common climbing fiber input likely converge onto the same nuclear cells. Did the authors record complex spikes? If so, pooling according to complex spike tuning should be attempted. If this pooling works the results become much stronger. The discussion section related to the model was vague and not convincing, particularly the appeals to energy efficiency. Is it not true that in the authors model the opposite responses in PCs to passive input are all cancelled in rFN? Surely this is not energy efficient. Numerous key issues are not discussed at all. Are synapses between PCs and rFN cells plastic? Is there a biologically plausible learning rule that could implement the weighting in the model? Some mention of what is or is not known about patterns of PC convergence onto rFN in macaques should be highlighted in the discussion. Again the issue of pooling based on climbing fiber input is highly relevant to the actual biological implementation.

In addition, I have a number of questions about the data and analysis in Figure 4 and 5.

-More details should be given about the Purkinje cell recordings. What fraction of total cells recorded behaved in the manner shown here. Were there differences depending on the lobule or mediolateral (zonal) location? Again, any data on complex spikes?

-I do not understand what is being shown in 4B. The cartoons suggest that the motor response should decrease in the left panel and increase on the right. I see the latter but not the former. The arrow in the cartoon points to the peak of the response to the motor command or active/passive modulation. What is the relevance of the resting rate on the x-axis? i.e. is there a reason not to compare the timecourse of the modulations (something closer to what is depicted in the cartoons)?

-What is the justification in Fig 5b,c for plotting only the direction that had the best fit? When all the data is shown (extended data fig 5) the trends are much less clear or absent. This seems troubling but maybe I am missing something here? How does this analysis decision connect to the modeling? Is all the data being used for the model or only the data plotted in the main figure? Zooming back out, is it even relevant if individual PCs exhibit cancellation on a cell by cell basis given how the model suggests the cancellation arises in the rFN?

Reviewer #3 (Remarks to the Author):

Summary

Cancelling the sensory consequences of our own movements is an important aspect of perception and motor control. In the vestibular system, vestibular nuclei neurons have been shown to distinguish between self-generated and unexpected, passive self-motion even though the vestibular afferents don't. In this paper, the authors test for the well-justified possibility that Purkinje cells in the anterior vermis may provide the signals necessary to suppress the expected, self-generated vestibular signals in the vestibular nuclei. To do so, they recorded from Purkinje cells under five conditions: passive rotation of the whole body, of only the body while the head remained stationary, when the animal actively moved its head, when the active head movement was combined with passive whole-body movement, and a condition in which the head was unexpectedly restrained when the animal attempted a head movement.

They found that Purkinje cells' sensitivity was reduced to active movements relative to passively generated movements, both when the two types of movements were independent or combined. Additionally, they found that the cells were responsive to attempted movements, a signal that could be used to explain the attenuation of the responses to active vs. passive movements. Finally, the authors use a model simulation to show that a weighted sum of heterogeneous Purkinje cell types can explain the firing rates of neurons in the recipient vestibular nucleus (the rostral fastigial nucleus).

Comments

Overall, this is an interesting study combining empirical and theoretical work to explain a key missing piece in the functioning of the vestibular system. I particularly appreciated how the authors provide an intuitive explanation of their theoretical predictions for how the possible convergence patterns of Purkinje cells could explain the activity of fastigial neurons in Figure 6A. My comments and questions are listed below:

1. If the main claim is that the attenuation of Purkinje cells to the active conditions drives the attenuation observed in the rostral fastigial nucleus, it is important to show that the attenuation observed in Purkinje cells precedes the attenuation observed in the rFN. Although the recordings may not be in the same cells or same animals, comparing a latency distribution in the current data with that of the data in Brooks and Cullen, 2009 would be informative.

2. What was the rationale for running the model in Figure 7 on two datasets – one including the attempted head movement condition and one without? Relatedly, in 7B, although the confidence

intervals overlap, the model without the attempted head movement condition appears to do better on average. Isn't this counter-intuitive given the prediction that the attenuation of active condition responses in the rFN is the output of a learned internal model? Is the proposal instead that the integration of the motor command and sensory input taking place upstream of or at the level of Purkinje cells at most such that only the outputs of this integration are inherited by rFN neurons?

3. It would be helpful if an early figure in the paper included a schematic of all the movement conditions in the experiment. It was a bit confusing to have to jump between sections of the paper to collect the overall study design.

In general, several points of the Methods should include a more detailed description in the interest of reproducibility:

4. Were all the conditions run on all the 63 neurons recorded for this study? There was a sentence in the Methods section which suggested that only a subset were recorded in the attempted head movement condition. It would be helpful to have a clear statement early in the Methods or Results of the number of neurons recorded in each condition. Similarly, it'd also be helpful to know the minimal inclusion criteria (in terms of the number of "trials" of each condition) for determining the sensitivity of the neurons.

5. All the figures with empirical data or their legends should include the number of neurons (n) used for the analysis.

6. How was the animal's head unexpectedly restrained in the attempted head movement condition? Was the implanted titanium headpost used in this case as well? If so, how was the timing of the restraint matched to the attempted head movement?

7. For the population models reported in Figure 7 and Extended Figures 6-11:

a. Was the model set up to predict rFN firing rates within specific bins and account for the dynamics of firing rates relative to the movements or averaged across the entire movement in two directions?

b. Presumably, the firing rates of Purkinje cells in each movement condition was used to predict rFN firing rates in that same condition (as schematized in Figure 7A). Was the optimization of the weights performed independently for each condition or assumed to have the same pattern across all conditions?

c. What was the generative model used to simulate additional Purkinje cells and which parameters of the empirical data did it preserve? If the model optimized for average firing rates across the entire movement, was it important to preserve the dynamics observed in the recorded data?

REVIEWER COMMENTS

Reviewer #1 (Remarks to the Author):

General comments

In the last sentence of the Introduction, the authors stated “Thus, taken together our results constitute the first direct demonstration of the widely held view that the cerebellum integrates movement-based predictions with actual sensory feedback to cancel the sensory consequences of active self-motion.” Unfortunately, I am not yet convinced as summarized below.

We thank the reviewer for his/her feedback and have addressed each of the reviewer’s specific concerns point by point below.

Specific comments

Major:

1. The presentation of this manuscript appears to depend too heavily on the two examples of Purkinje cells (PCs) (Cell 1 and Cell 2) in Figs. 1, 3, 4 and 5. If you used different neurons for different figures, you should mention it in the figure legends.

We appreciate that all reported analyses should be based on population data. We illustrated these specific example cells to demonstrate examples of our main findings. We have revised the manuscript to denote which specific example cells are illustrated in Figs. 1, 3, 4, and 5 in the scatter plots corresponding to the population results, and now mention this in the figure legends. Specifically, Fig.2C (for the cells in Figure 1), Figure 3B & 3C (for the cells in 3A), Figure 4B (for the cells in 4A), and Figure 5B & 5C (for the cells 5A). We also note that we illustrated data from different neurons and have corrected an error in the revised text left from an earlier draft regarding neurons shown in fig. 3 versus 1. Additionally, in the revised manuscript, to address the reviewer’s concern we now clarified on page 7 when group and individual statistics were performed.

2. The authors modeled PC activities as a linear sum of activities proportional to the position, the velocity, and the acceleration (page 18, Equation 1). Nevertheless, almost the entire analysis in this manuscript is somehow dedicated to analyses of velocity-related PC activities. Internal models are not dedicated to the prediction of velocity.

All values that we reported were based on the computation of the coefficients for all three dynamic terms (i.e., position, velocity, and acceleration) as was detailed in Equations 2 and 3. We then report the sensitivity and phase *relative* to velocity, such that the acceleration and position coefficients contribute to the phase lead and lag, respectively. We note that we chose to represent phase relative to velocity because since velocity is the dominant term. To address the reviewer’s comment and make this point directly we have added a new figure that demonstrates these results (i.e., panel B of the new Extended Data Figure 1). Additionally, we have revised Figure 1 to include head position as well as velocity traces to clarify that our dynamic analysis did not focus only on velocity.

3. The authors raised important points in the Introduction as follows -(1)(2)(3):

(1) Page 3, lines 14-17: “The vestibular afferents of the VIII nerve encode sensory signals in a manner that is not dependent on context; their responses are comparable whether self-motion is unexpected or intentional”.

(2) Page 3, lines 17-21: “In contrast, the vestibulospinal pathway neurons within the vestibular

nuclei, which are directly targeted by these same afferents, selectively respond to vestibular sensory stimulation during unexpected (i.e., passive) self-motion, as do neurons in the ascending posterior thalamocortical vestibular pathway. This selective encoding of unexpected vestibular information provides a neural mechanism for ensuring postural and perceptual stability during self-motion”.

(3) Page 3, lines 24-28: “To date, however, the source of the suppression signal that cancels peripheral vestibular input to central pathways during expected (i.e., active) self-motion remains unknown. Importantly, vestibular nuclei neurons are capable of distinguishing between expected and unexpected vestibular stimuli, even when these stimuli occur simultaneously”.

Based on (1), (2), (3), and because the vestibular nuclei receive major inputs from the VIII nerve and PCs in the cerebellar cortex, it is most likely that the suppression signal comes from the cerebellar cortex. Namely, the PCs send it to the vestibular neurons. This inference predicts that PC activities encode the motor command's sensory consequences, rather than simple attenuation of sensory responses as demonstrated in Figs. 1, 2, and 3. This inference also predicts that PC activities represent the negative of the canceling signal because PCs are inhibitory. Nevertheless, neither prediction does not appear to match the results of this manuscript.

We agree with the reviewer that Purkinje cells likely suppress vestibular neuron signals. Indeed, our paper's central thesis is that collective response of Purkinje cells, rather than individual cells, is required for cancellation.

-First, our results in Figs. 1-5 show that individual Purkinje cells integrate sensory and motor information in such a way that they respond differently to active and passive motion.

Importantly, not only are responses to active motion are suppressed, the direction of the response flips for ~50% of Purkinje cells.

-Secondly, while responses of individual Purkinje cells cannot account for the generation of a suppression signal, the heterogeneity and notably changes in response direction (see theoretical model in Figure 6 and Extended Data Figure 7), allows the collective Purkinje cell activity to encode sensory consequences and create a 'negative image' of the canceling signal. Our new additional modeling, in which we now added the VIII nerve inputs to the vestibular nuclei and their interaction with the Purkinje cells' input, highlights the distinction between individual and collective Purkinje cell responses. We also now more directly emphasize these points in our presentation of Figure 7 (Page 10).

4. Page 2, lines 9-10, Abstract: “Purkinje cells exhibit selective suppression for actively versus passively generated vestibular input, even when passive stimulation is experienced simultaneously.”: This result appears to suggest that the source of the suppression signals mentioned above are generated before PCs. Then where? This issue should be explained and discussed.

We thank the reviewer for this comment. We have clarified this point in our revised manuscript (Page 8), namely that Purkinje cells' suppression during active movements does not imply suppression precedes their activation. Our results show that individual Purkinje cells integrate motor and sensory information, and that this suppression can arise from the pooled response of a population Purkinje cells themselves (as shown in Figure 6).

5. PCs are GABAergic and exert tonic inhibition on vestibular nuclei neurons. Therefore, the suppression or decrease of “actively generated vestibular” response in PCs should result in disinhibition (i.e., facilitation) of target vestibular nuclei neurons as demonstrated before

(Ishikawa et al., 2014, PLoS ONE, 9(10): e108774). It appears that this paper has not factored in this basis of cerebellar physiology.

As noted in response to comment #3 above, we agree that the inhibitory role of Purkinje cells on deep cerebellar and vestibular nuclei makes our single-cell level results initially counterintuitive. To address the reviewer's comment and make our findings more intuitive we have performed new additional modeling that now includes VIII nerve input in revised Figure 7. Importantly, these new modeling results directly demonstrate that this input can be cancelled by the collective input of a population of Purkinje cells. We have also revised our manuscript to better emphasize two key findings: first, individual Purkinje cells exhibit selective suppression of signals related to active, as opposed to passive, vestibular inputs; second, our simplified model successfully accounts for this selective suppression by incorporating the inhibitory projections from Purkinje cells to the nuclei, which is detailed in Figures 6 and 7 (Page 9).

6. Page 7, lines 27-30: "In the active condition, in addition to receiving sensory feedback, the brain also generates motor commands to activate the neck musculature. Accordingly, we hypothesized that the motor-related inputs to the anterior vermis might account for the reduced sensitivity of Purkinje cells to active versus passive head motion". I agree that this is a possibility. Then where can we find the evidence of the efference copy activities of PCs in this manuscript?

Evidence that Purkinje cells encode motor-related signals was provided in Figure 4 of the original manuscript. For the experiment shown in this figure, the head was completely restrained so that there was no head motion. Neurons then exhibited a response to the generation of a motor command during attempted head movement, as indicated by the presence of neck torque. To be more accurate in describing our findings, the text and figure labels to emphasize the evidence for "motor-related" signals, rather than using the term "efference copy" of the motor command (pages 17 and 19).

7. Fig. 4, the legend: "The monkey's head was unexpectedly restrained as it oriented to an eccentric target, where the intended but unrealized movement was demonstrated by the production of neck torque".

I cannot agree to assume the neck torque (blue lines in Fig. 4) as "attempted" because the gradually incremental torque profile (Fig. 4) must be different from a torque profile for a natural neck movement without a restraint (e.g., Fig. 1). I would recommend the term "perturbed neck torque". Note that the temporal patterns of PC activities (gray areas) do not match with those of neck torque (blue lines).

We disagree with the reviewer given that the profiles depicted actually represent a sudden change in torque occurring within a 100 ms timeframe, not a gradual increase, as a result of the short timescale involved. To make this point clearer, the revised text now highlights this rapid change from 0-50 within 100 ms and specifies that our analysis targets the initial movement phase (50ms), isolating responses prior to the activation of long-latency stretch reflexes or voluntary adjustments, as described in Maeda et al., 2020, Current Biology (page 8).

Furthermore, there is no inherent reason why the dynamics of PC activities should directly mirror neck torque. Our previous work, "Zobeiri and Cullen 2022," demonstrates that the same Purkinje cells show diverse dynamics in response to vestibular and proprioceptive stimuli. This diversity similarly extends to their torque response dynamics, a point we have made more explicit in the revised manuscript (Page 8).

8. The authors may want to compare the present study with a recent study that has provided neural evidence of the cerebellar forward model in the cerebrocerebellum (Tanaka et al. (1), (2)). The cerebrocerebellum, forming a closed loop between cortical motor areas, receives both the efference copy and somatosensory inputs required for a neuronal substrate to serve as a forward model. They recorded neural activities in monkeys performing step-tracking movements. The activities of mossy fibers (MFs) at time $t+t1$ (the neural activity encoding a future state) were well reconstructed as a weighted sum of dentate nucleus cells (DNCs) at time t (the neural activity representing the current output of the cerebellum). This analysis provided the first direct evidence for the internal forward model hypothesis of the cerebrocerebellum. In addition, they established a set of linear equations to relate the activities of the cerebellar neurons. First, the PC activities were reconstructed linearly as the weighted sum of the MF activities. Second, the DNC activities were reconstructed linearly as the weighted sum of the PC activities and the MF activities. These linear equations were compatible with those of the optimal predictor known as the Kalman filter (Tanaka et al. (1), (2)). In the Kalman-filter framework, the predictive step computes the predictive state (PC activities) from the current state (MF activities), and the filtering step (DNC activities) integrates the predictive state (PC activities) and sensory feedback from the periphery (another set of MF activities).

(1) Tanaka et al. (2019) Neural evidence of the cerebellum as a state predictor. *Cerebellum*. 18, 349-371.

(2) Tanaka et al. (2020) The cerebro-cerebellum as a locus of forward model: a review. *Front Sys Neurosci*. 14:19.

We thank the reviewer for this suggestion. We have revised our Discussion to include a discussion of the Tanaka et al., paper findings regarding the lateral hemispheres of the cerebellar cortex (i.e., cerebero-cerebellum) (page 14). As noted by the reviewer, the findings of these papers focused on neural recordings during voluntary wrist movements in monkeys are relevant to our study. First, we note that the finding that the firing rates of deep cerebellar nuclei neurons can be reconstructed as a weighted linear sum of cerebero-cerebellum Purkinje cell (and Mossy fiber) responses – similar to our present findings. This is interesting since this region of the cerebellum also likely integrates sensory and motor inputs; it receives somatosensory input which it is presumed is integrated with presumed motor inputs. We also discuss (page 14) the authors conclusion that they can predict mossy fibers responses as the weighted sum of deep cerebellar nuclei activity in the preceding trial, in a manner compatible with a Kalman filter (Tanaka et al. 2019, 2020) in the context of our present results. Testing whether similar mechanism function within our system could provide a promising avenue for future research into forward models in the cerebellum, given that the suppression of vestibular reafference in our system represents direct evidence of a computation reliant on the internal forward model.

9. In Fig. 3, the phase lags (i.e., delays) between the passive head velocity and PC activities appear significantly different for PC1 and PC2. I would expect consistent delays for reliable prediction. How do you explain the variation?

As we indicated in the text (page 6) and Extended data figure 5, there is a heterogeneity in the responses of Purkinje cells, showing a diverse range of phase delays with the response lead or large the movement. In this context it is expected that there would be differences in the phase lags across different example cells. This diversity wouldn't affect the reliable predictions as the phase lag was modeled for each cell separately. Additionally, in case that the visual representation of head velocities creates confusion, we depict the velocity traces with the

opposite sign, as indicated in the figure legend, to simplify the comparison between head velocity and firing rate.

Specific comments

Minor:

1. There is little description of how the example PCs (in Figs. 1, 3, 4, and 5) are representative of the whole population of PCs recorded in this study. It is helpful to identify these representative neurons in plots of Figs 2ABC, 3BC, 4B, and 5BC.

As noted in our response to Major Comment 1, we appreciate the concern and now identify the example PCs in the scatter plots showing the corresponding population results.

2. The location of the PCs in the cerebellar cortex is essential information to be presented as an inset.

In the methods section, page 16, we now report the location of the Purkinje cells, specifically “Histological analysis confirmed that the Purkinje cells were located in lobules II–V of the anterior vermis, ~0–2 mm from the midline” We note the location of these the Purkinje cells was the same as for those neurons recently described in Zobeiri and Cullen 2022.

3. For Figs. 3, 4, and 5, it is necessary to describe calculated values for the example neurons.

As noted in our response to Major Comment 1, we have revised the figures and associated text so that the example Purkinje cells are now identified in the scatter plots, such that the calculated values can be compared to those of the population of neurons recoded in our study.

4. In Fig. 4A, PC1, the onset of suppression (left) appears too late for the torque change. In addition, the correlation of their temporal patterns is poor.

As noted in our response to Major Comment 7, we have shown previously these same Purkinje cells show considerable heterogeneity in the response dynamics to sensory vestibular and proprioceptive inputs (i.e. Zobeiri and Cullen 2022). They likewise appear to demonstrate heterogeneity in their response dynamics to torque. We have revised the text to emphasize this point.

5. In Fig. 4, where is the torque onset? The hatched lines in the center? However, the torque level is already significantly deviated from the baseline at the center.

We appreciate the reviewer's comment and have revised the figure and its associated legend to improve clarity. In particular, we clarify that we defined the torque onset as the point where the torque value exceeds 2 standard deviations from the torque values at rest. We have updated the figure to depict this range of variability (x2 SD).

Reviewer #2 (Remarks to the Author):

The ms by Zobeiri and Cullen is the latest in a series investigating vestibular coding during active and passive self-motion. Elegant prior work by the same group has shown that neurons in the rostral fastigial nucleus (rFN) of the cerebellum (Brooks and Cullen, 2013, 2015), as well as neurons in the vestibular nucleus, selectively encode passive head rotation while remaining insensitive to vestibular input when it is caused by the animal's own movement. The hypothesis from prior work is that motor-based predictions cancel the self-generated input, however, the important questions of where/how the cancellation signal is constructed remain unclear. The present study extends the investigation from the output stage of the cerebellum (rFN neurons) to Purkinje cells of the anterior vermis that are presumed to inhibit rFN cells. Response properties to active and passive motion regimes tested previously in rFN and vestibular nucleus are tested in Purkinje cells. A model of rFN responses is proposed in which cancellation of vestibular input during active head rotation can be accounted for based on a weighted sum of ~40 Purkinje cell inputs.

This is a logical and important next step in the authors' investigations with broad potential relevance to understanding cerebellar function and sensorimotor processing. However, I have several major concerns.

We thank the reviewer for his/her feedback and have addressed each of the reviewer's specific concerns point by point below.

1a) First, the connection to the closely related past work is poorly communicated, making it difficult to appreciate the new results being shown. The abstract states "Here, we provide the first direct evidence that the cerebellum combines incoming sensory signals with internally generated predictive motor signals to selectively cancel actively generated sensory input." I was confused by this, because the authors have already demonstrated adaptive cancellation of self-generated input in the cerebellar nucleus in, at least, two prior papers. The intro focuses on vestibular nucleus and barely mentions rFN or the fact that the anterior vermis PCs recorded here presumably inhibit rFN (but not VN presumably).

We have revised the abstract and introduction (page 3) to emphasize the novelty of our study and emphasize that cerebellar Purkinje cells inhibit rFN as well as VN neurons, which both demonstrate the suppression of self-generated input as mentioned by the reviewer more clearly. Specifically, our goal was to determine the source of this cancellation, and our study makes two important contributions. First, we show that anterior vermis Purkinje cells differentially encode active versus passive movements in a manner consistent with their motor-related inputs. This contrasts with rFH and VN neurons, which show no modulation to the generation of head motor commands. Second, we further show, via our modeling results in Figure 6 and new additional modeling in Figure 7, incorporating direct vestibular inputs, that pooling the responses of physiologically realistic population (i.e. ~40) of anterior vermis Purkinje cells can account for the cancellation observed at the level of the vestibular/cerebellar nuclei, consistent with it being the source of cancellation.

1b) Also, what about connections between rFN and VN? Although the significance here lies in understanding how the cancellation in rFN is done (rather than in understanding PC responses in isolation), this is not communicated clearly enough until the final section of the results. Does the above sentence from the abstract refer specifically to the finding that motor signals in Purkinje cells appear to partially cancel self-generated input? If so, this raises the question of

whether this “computation” at the level of the PCs actually supports the cancellation at the output stage. This is far from clear.

We thank the reviewer for this comment and have revised the text to provide a clearer explanation of this circuit (page 3). Because VN neurons like rFN neurons, receive direct inputs from Purkinje cells, the VN effectively functions as a fourth deep cerebellar nucleus (Goldberg et al., 2012). In addition, VN and rFN neurons are reciprocally connected (Noda et al. 1990), and our prior work has demonstrated that neurons both areas demonstrate comparable cancellation for reafferent vestibular inputs during active versus passive head movements (e.g., 70%).

1c) Since PCs are inhibitory, they would be expected to transmit the motor signals to rFN cells instead of cancelling them out (as the authors point out). With appropriate weighting, the passive responses of different Purkinje cells can be averaged out, allowing the residual responses to self-generated motion to serve as the cancellation signal. However, in this scheme it is unclear why cancellation at the level of PCs needs to happen in the first place. Framing what happens in individual PCs as an elegant computation when it needs to be immediately undone by pooling populations at the level of the nucleus is confusing and distracts from the actual biological findings which are valuable even though they may be nuanced/complex.

We appreciate the comment- but note that the change in Purkinje Cell responses is more complex than just “cancellation”. Indeed, during active movements, not only do Purkinje Cells i) remain responsive but ii) the sensitivity of ~50% of neurons actually flips to change direction (Figure 2C). This flipping of the response directionality markedly contrasts with what we have previously observed in either the VN or rFN. In these target structures, the reduced neural responses retain the same directionality. Importantly, our model, depicted in Figure 6, demonstrates that this “flipping” characteristic of anterior vermis Purkinje cells is essential to generate a cancellation signal. To address the reviewer’s concern, we have the text to emphasize this point (page 9).

2a) Second, the modeling and related discussion can be expanded upon and improved. The authors should attempt to fit other aspects of rFN responses that they have previously recorded in Brooks and Cullen 2013, 2015. For example, can the same weighted population of PCs explain the prior finding that rFN neurons don’t respond to attempted head movements? I believe the authors have shown previously that cancellation does not occur if the motor command does not match the proprioceptive input. Can the model explain this?

We had addressed this point in the initial submission (original Figure 7, orange traces indicate results that included the attempt head movement condition) but appreciate that we could have been clearer. Thus, in the revised manuscript, we have moved the results of this simulation to a new figure (Extended Data Figure 9) and have expanded the accompanying text (page 10) to describe our approach/results in more detail. Specifically, we found that although Purkinje cells respond to motor-related inputs, the convergence of the population of 40-50 Purkinje cells can cancel out the effect of these inputs on their target neurons, such that their target neurons don’t respond to attempted head movements (Extended Data Figure 9, panel C right).

Prior work in oculomotor vermis has shown that summing PC responses pooled based on complex spike sensitivity input leads to useful signals (work from Herzfeld and Shadmehr should be cited). This makes sense because PCs that share common climbing fiber input likely converge onto the same nuclear cells. Did the authors record complex spikes? If so, pooling according to complex spike tuning should be attempted. If this pooling works the results become much stronger. The discussion section related to the model was vague and not convincing,

particularly the appeals to energy efficiency. Is it not true that in the authors model the opposite responses in PCs to passive input are all cancelled in rFN? Surely this is not energy efficient.

We agree with the reviewer that understanding whether PC response can be pooled based on complex spike activity is very interesting. While no prior studies have reported on the activity of complex spikes in the vestibular part of the anterior vermis, our preliminary analysis suggests that they do not consistently align with directional sensitivity. As noted in the revised Discussion (page 14), we anticipate that the development of such models will be an important but long endeavor and will therefore be the subject of future studies.

Numerous key issues are not discussed at all. Are synapses between PCs and rFN cells plastic? Is there a biologically plausible learning rule that could implement the weighting in the model?

We appreciate the reviewer's comment, but note that our task was not a learning task. As such the weights in our modeling were held consistent across conditions. To clarify this point, we have revised the Discussion to emphasize that we modelled our data across conditions with one set of synaptic weights. Differences across conditions, thus relate to differences in the inputs available to the network, specifically motor-related inputs. As noted in the revised Discussion, we believe that understanding how these synaptic weights change during a considered task (such as that conducted by Brooks et al. in 2016) is very interesting direction for future work (page 14).

Some mention of what is or is not known about patterns of PC convergence onto rFN in macaques should be highlighted in the discussion. Again the issue of pooling based on climbing fiber input is highly relevant to the actual biological implementation.

We have revised the text to specific that we are not aware of any specific studies that have described the patterns of anterior vermis PC convergence onto rFN in macaques. Likewise, as noted above, no prior studies have described complex spikes to head motion stimulation in the anterior vermis and we agree it is very interesting area for further work (page 14).

In addition, I have a number of questions about the data and analysis in Figure 4 and 5

-More details should be given about the Purkinje cell recordings. What fraction of total cells recorded behaved in the manner shown here. Were there differences depending on the lobule or mediolateral (zonal) location? Again, any data on complex spikes?

We have revised the methods to specify that we restricted our analysis to those neurons in the anterior vermis that responded to passive horizontal axis vestibular stimulation and estimate that this was ~30% of the neurons we encountered during our recording sessions. Of these neurons, ~75% were sensitive to passive proprioceptive stimulation comparable to the percentage reported in Zobeiri and Cullen 2022. Additionally, as was mentioned in the results of the original submission (Figure 4) "we found that the majority of Purkinje cells in our population (80%) responded during the production of a neck torque in at least one direction." And as we responded to the related comment above, we agree that the analysis of climbing fibers will be an interesting area for further work.

-I do not understand what is being shown in 4B. The cartoons suggest that the motor response should decrease in the left panel and increase on the right. I see the latter but not the former. The arrow in the cartoon points to the peak of the response to the motor command or

active/passive modulation. What is the relevance of the resting rate on the x-axis? i.e. is there a reason not to compare the timecourse of the modulations (something closer to what is depicted in the cartoons)?

We appreciate the reviewer's comment and have revised the cartoons in this figure and corresponding text (page 8) to more clearly demonstrate the importance of the resting rate. Specifically, the deviation of the firing rate due to the generation of head torque and the resting rate (represented by the deviation from unity line in the scatter plots) indicates the responsiveness of the Purkinje cells to motor-related input.

-What is the justification in Fig 5b,c for plotting only the direction that had the best fit? When all the data is shown (extended data fig 5) the trends are much less clear or absent. This seems troubling but maybe I am missing something here?

To address the reviewer's comment, we have now revised Figure 5 and corresponding text in page 8 (and its corresponding supplementary figure, Extended Data Figure 6) to illustrate the direction with the largest response to the motor-related input, as measured by the change from resting rate. We had elected to show the direction with the best fit to demonstrate that the summation model can provide an excellent prediction of a given neurons responses in at least one direction across our population of Purkinje cells.

How does this analysis decision connect to the modeling? Is all the data being used for the model or only the data plotted in the main figure? Zooming back out, is it even relevant if individual PCs exhibit cancellation on a cell by cell basis given how the model suggests the cancellation arises in the rFN?

We have clarified in the text in page 10 that we our population modeling included the responses of all neurons, in the noted conditions, in both directions. Specifically, we first estimated our model using the 4 conditions (passive whole-body, body-under-head, and passive and active head-on-body), for which we had data from all neurons in our population – for movements in both directions. Then, we compared these results to those obtained when we included a 5 condition, namely attempted head movement- population – again for movements in both directions. Regarding the generation of the cancellation signal from a population of Purkinje cells, please refer to our response to the Major comment 1a above.

Reviewer #3 (Remarks to the Author):

Summary

Cancelling the sensory consequences of our own movements is an important aspect of perception and motor control. In the vestibular system, vestibular nuclei neurons have been shown to distinguish between self-generated and unexpected, passive self-motion even though the vestibular afferents don't. In this paper, the authors test for the well-justified possibility that Purkinje cells in the anterior vermis may provide the signals necessary to suppress the expected, self-generated vestibular signals in the vestibular nuclei. To do so, they recorded from Purkinje cells under five conditions: passive rotation of the whole body, of only the body while the head remained stationary, when the animal actively moved its head, when the active head movement was combined with passive whole-body movement, and a condition in which the head was unexpectedly restrained when the animal attempted a head movement.

They found that Purkinje cells' sensitivity was reduced to active movements relative to passively generated movements, both when the two types of movements were independent or combined. Additionally, they found that the cells were responsive to attempted movements, a signal that could be used to explain the attenuation of the responses to active vs. passive movements. Finally, the authors use a model simulation to show that a weighted sum of heterogenous Purkinje cell types can explain the firing rates of neurons in the recipient vestibular nucleus (the rostral fastigial nucleus).

Comments

Overall, this is an interesting study combining empirical and theoretical work to explain a key missing piece in the functioning of the vestibular system. I particularly appreciated how the authors provide an intuitive explanation of their theoretical predictions for how the possible convergence patterns of Purkinje cells could explain the activity of fastigial neurons in Figure 6A. My comments and questions are listed below:

We thank the reviewer for his/her positive feedback and have addressed each of the reviewer's specific concerns point by point below.

1. If the main claim is that the attenuation of Purkinje cells to the active conditions drives the attenuation observed in the rostral fastigial nucleus, it is important to show that the attenuation observed in Purkinje cells precedes the attenuation observed in the rFN. Although the recordings may not be in the same cells or same animals, comparing a latency distribution in the current data with that of the data in Brooks and Cullen, 2009 would be informative.

We appreciate the reviewer's comment. To directly address this point, we have conducted a new analysis that compute the average latency of when the response in the active condition is suppressed relatively to the passive condition (panel D of Supplementary Figure 3). We found that, on average, the timing if this difference preceded head motion by ~20 ms. Given that the rFN is only 1 synapse away from the Purkinje cells, however, it would not be possible to detect a timing difference at this next stage given the variability of this measure. Importantly, the value we estimate in our new analysis is comparable with the observed timing of the motor signal in the Purkinje cell (Fig. 5D).

2. What was the rationale for running the model in Figure 7 on two datasets – one including the attempted head movement condition and one without? Relatedly, in 7B, although the confidence

intervals overlap, the model without the attempted head movement condition appears to do better on average. Isn't this counter-intuitive given the prediction that the attenuation of active condition responses in the rFN is the output of a learned internal model? Is the proposal instead that the integration of the motor command and sensory input taking place upstream of or at the level of Purkinje cells at most such that only the outputs of this integration are inherited by rFN neurons?

We appreciate the reviewer's comment and have revised the text to clarify our approach (page 10). Specifically, we first estimated our model using the 4 conditions (passive whole-body, body-under-head, and passive and active head-on-body), for which data from all neurons in our population was included in the model. In our modelling, we then next estimated our model including a 5th condition, namely attempted head movement. Since we had data from fewer neurons in this condition, we used augmentation (please see revised Methods) to model the additional 5th condition. Given that we performed our modeling using data from all conditions simultaneously, adding more conditions imposes additional criteria on the model, and so as expected this resulted in reduced model performance.

Finally, in response to the reviewers last comment, we emphasize that the attenuation of the rFN/VN neurons indeed provides compelling evidence for the *existence* of an internal forward model. However, the response of these neurons is not consistent with being the *source* of the internal model because they do not receive motor-related inputs. In this context, our current findings provide evidence for the source of the internal model, which generates the sensory consequences of the motor command, namely a population of Purkinje cells; these cells receive motor-related input and integrate it with sensory information.

3. It would be helpful if an early figure in the paper included a schematic of all the movement conditions in the experiment. It was a bit confusing to have to jump between sections of the paper to collect the overall study design.

We agree with the reviewer and have added a new Supplementary Figure 1A to the manuscript which illustrates all of the movement conditions.

In general, several points of the Methods should include a more detailed description in the interest of reproducibility:

4. Were all the conditions run on all the 63 neurons recorded for this study? There was a sentence in the Methods section which suggested that only a subset were recorded in the attempted head movement condition. It would be helpful to have a clear statement early in the Methods or Results of the number of neurons recorded in each condition. Similarly, it'd also be helpful to know the minimal inclusion criteria (in terms of the number of "trials" of each condition) for determining the sensitivity of the neurons.

The reviewer is current that we recorded all 63 neurons during our 4 active and passive head-on-body movements conditions. Of these neurons, a subset of 34 Purkinje cells were also tested during attempted head movements, and a subset of 31 Purkinje cells were also tested in the combined passive-active condition. To ensure robustness in our quantification of responses, we only included cells for which we had data for at least 10 head movements in each direction. The Methods section has been revised to include these inclusion criteria (page 16).

5. All the figures with empirical data or their legends should include the number of neurons (n) used for the analysis.

We have revised the figure legends and now include the number of neurons used in each analysis.

6. How was the animal's head unexpectedly restrained in the attempted head movement condition? Was the implanted titanium headpost used in this case as well? If so, how was the timing of the restraint matched to the attempted head movement?

We appreciate the reviewer's comment and have modified the Methods section to more clearly explain this experiment (page 17). Specifically, we applied random brakes to the head movements after the monkey had oriented to a target for >500ms, just prior (100ms) to presenting the next target. To do this, we activated an electromagnetic clutch (Placid Industries), attached to the head-holder. Such breaks were applied unexpectedly, in head-unrestrained monkeys, for a small subset of trials (less than 5%). We then measured the torque while the monkey, without knowledge of the imposed restraint, tried to make head movements.

7. For the population models reported in Figure 7 and Extended Figures 6-11:

a. Was the model set up to predict rFN firing rates within specific bins and account for the dynamics of firing rates relative to the movements or averaged across the entire movement in two directions?

The reviewer is correct. We considered the dynamics of firing rates in relation to the movements for the entire movement (sampled at 1kHz) for two directions, and now include this information in the revised method section. (page 20)

b. Presumably, the firing rates of Purkinje cells in each movement condition was used to predict rFN firing rates in that same condition (as schematized in Figure 7A). Was the optimization of the weights performed independently for each condition or assumed to have the same pattern across all conditions?

The optimization of the weights was computed across all conditions. We have revised the methods and results sections to clarify this point. Page 20)

c. What was the generative model used to simulate additional Purkinje cells and which parameters of the empirical data did it preserve? If the model optimized for average firing rates across the entire movement, was it important to preserve the dynamics observed in the recorded data?

The generative model used to simulate additional Purkinje cells for the attempted head movement condition utilizes parameters that reproduced the dynamics of empirical data. We have revised the methods and results sections to make this point clearer (page 20)

REVIEWERS' COMMENTS

Reviewer #1 (Remarks to the Author):

Congratulations! I am happy with this revised manuscript.

Reviewer #2 (Remarks to the Author):

The authors have addressed my major concerns.

Reviewer #3 (Remarks to the Author):

The authors have satisfactorily addressed my concerns in their revision. I found their responses and the updated clarifying text in the manuscript to be helpful.

Reviewer #1:

Congratulations! I am happy with this revised manuscript.

Reviewer #2:

The authors have addressed my major concerns.

Reviewer #3:

The authors have satisfactorily addressed my concerns in their revision. I found their responses and the updated clarifying text in the manuscript to be helpful.

We thank the Reviewers for their encouraging feedback. We also thank the Editors and Reviewers for their comments and suggestions, which have served to improve our manuscript.